# Landmark-Guided Subgoal Generation in Hierarchical Reinforcement Learning

**Junsu Kim**[1]     **Younggyo Seo**[1]     **Jinwoo Shin**[1,2]

[1]Kim Jaechul Graduate School of AI     [2]School of Electrical Engineering
Korea Advanced Institute of Science and Technology (KAIST)
{junsu.kim, younggyo.seo, jinwoos}@kaist.ac.kr

## Abstract

Goal-conditioned hierarchical reinforcement learning (HRL) has shown promising results for solving complex and long-horizon RL tasks. However, the action space of high-level policy in the goal-conditioned HRL is often large, so it results in poor exploration, leading to inefficiency in training. In this paper, we present HIerarchical reinforcement learning Guided by Landmarks (HIGL), a novel framework for training a high-level policy with a reduced action space guided by *landmarks*, i.e., promising states to explore. The key component of HIGL is twofold: (a) sampling landmarks that are informative for exploration and (b) encouraging the high-level policy to generate a subgoal towards a selected landmark. For (a), we consider two criteria: coverage of the entire visited state space (i.e., dispersion of states) and novelty of states (i.e., prediction error of a state). For (b), we select a landmark as the very first landmark in the shortest path in a graph whose nodes are landmarks. Our experiments demonstrate that our framework outperforms prior-arts across a variety of control tasks, thanks to efficient exploration guided by landmarks.[1]

## 1   Introduction

Deep reinforcement learning (RL) has demonstrated wide success in a variety of sequential decision-making problems, i.e., board games [38, 42], video games [1, 26, 38], and robotic control tasks [15, 30, 53]. However, solving complex and long-horizon tasks has still remained a major challenge in RL, where hierarchical reinforcement learning (HRL) provides a promising direction by enabling control at multiple time scales via a hierarchical structure. Among HRL frameworks, goal-conditioned HRL has long been recognized as an effective paradigm [5, 20, 29, 37], showing significant success in a variety of long and complex tasks, e.g., navigation with locomotion [21, 54]. The framework comprises a high-level policy and a low-level policy; the former breaks the original task into a series of subgoals, and the latter aims to reach those subgoals.

The effectiveness of goal-conditioned HRL depends on the acquisition of effective and semantically meaningful subgoals. To this end, several strategies have been proposed, e.g., learning the subgoal representation space [6, 11, 21, 29, 31, 32, 35, 44, 50] or utilizing domain-specific knowledge for pre-defining subgoal space [28, 54]. However, the learned or pre-defined subgoal space is often too large, which results in poor high-level exploration, leading to inefficient training. To address this issue, Zhang et al. [54] recently proposed a reduction of the high-level action space into the $k$-step adjacency region around the current state. This approach, however, is limited in that it considers all the states within the $k$-step adjacency region equally as the candidate actions for the high-level policy, without considering the *novelty* of states, which is crucial for exploration.

---

[1]Code is available https://github.com/junsu-kim97/HIGL

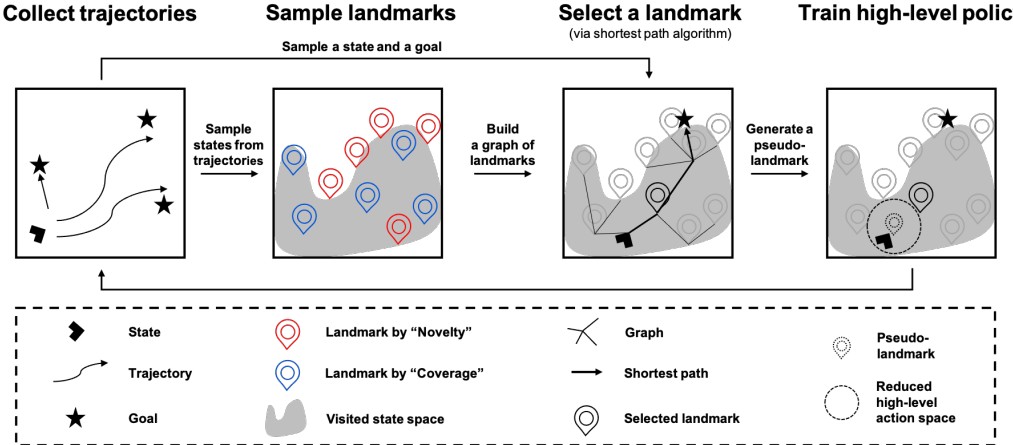

Figure 1: Illustration of **HI**erarchical reinforcement learning **G**uided by **L**andmarks (HIGL). (1) We collect trajectories using a high-level and a low-level policy. (2) Sample landmarks from visited states based on "coverage" and "novelty" criteria, respectively, and merge them. (3) Select a single landmark among the sampled landmarks in a graph constructed by landmarks, a goal, and a current state. (i.e., select the very first landmark in the shortest path to the goal). (4) Train a high-level policy to generate a subgoal toward the selected landmark.

**Contribution.**    In this paper, we present **HI**erarchical reinforcement learning **G**uided by **L**andmarks (HIGL), a novel framework for training a high-level policy that generates a subgoal toward *landmarks*, i.e., promising states to explore. HIGL consists of the following key ingredients (see Figure 1):

- **Landmark sampling:** To effectively sample landmarks that represent promising states to explore, it is important to sample landmarks that cover a wide area of state space and contain novel states. To this end, we propose two sampling schemes: (a) coverage-based sampling scheme that samples located as far away from each other as possible and (b) novelty-based sampling scheme that stores novel states encountered during training and utilizes them as landmarks. We find that our method successfully samples diverse and novel landmarks.

- **Landmark-guided subgoal generation:** Among the sampled landmarks, we select the most *urgent* landmark by our landmark selection scheme with the shortest path planning algorithm. Then we propose to *shift* the action space of a high-level policy toward a selected landmark. Because HIGL constructs a high-level action space that is both (a) reachable from the current state and (b) shifted towards a promising landmark state, we find that the proposed method can effectively guide the subgoal generation of a high-level policy.

We demonstrate the effectiveness of HIGL on various long-horizon continuous control tasks based on MuJoCo simulator [48], which is widely used in the HRL literature [9, 16, 28, 29]. In our experiments, HIGL significantly outperforms the prior state-of-the-art method, i.e., HRAC [54], especially in complex environments with sparse reward signals. For example, HIGL achieves the success rate of 65.1% in Ant Maze (sparse) environment, where HRAC only achieves 17.6%.

## 2   Related work

**Goal-conditioned HRL.**    By introducing a hierarchy consisting of a high-level policy and a low-level policy, goal-conditioned HRL has been successful in a wide range of tasks. Notably, Nachum et al. [28] proposed an off-policy correction method for goal-conditioned HRL, and Levy et al. [19] successfully trained multiple levels of policies in parallel with hindsight transitions. However, the acquisition of effective and semantically meaningful subgoals still remains a challenge. Several works have been proposed to address this problem, including learning-based approaches [6, 11, 21, 29, 31, 32, 35, 44, 50] and domain knowledge-based approaches [28, 54]. The work closest to ours is HRAC [54] that reduces the high-level action space to the $k$-adjacent region of the current state. Our work differs in that we explicitly consider the *novelty* of each state within the region instead of treating all states in an equal manner.

**Subgoal discovery.** Identifying useful subgoals to guide training (low-level) policy has long been recognized as an effective way to solve complex RL problems [23–25, 43]. Recent works provide a subgoal by (1) constructing an environmental graph and (2) planning in the graph [8, 14, 17, 22, 36, 41, 51, 52]. For (1), one important point is to build a graph that represents the entire map enough with a limited number of nodes. To this end, several approaches such as farthest point sampling [14] and sparsification [17] were proposed. For (2), traditional planning algorithms, e.g., the Bellman-Ford algorithm, are usually used for offering the most emergent node to reach the final goal. These works have shown promising results in complex RL tasks but have a scalability issue due to increasing planning time with a larger map. Meanwhile, our framework differs from this line of works since ours "train" high-level policy rather than hard-coded high-level planning to generate a subgoal.

## 3 Preliminaries

We formulate a control task with a finite-horizon, goal-conditioned Markov decision process (MDP) [45] defined as a tuple $(\mathcal{S}, \mathcal{G}, \mathcal{A}, p, r, \gamma, H)$, where $\mathcal{S}$ is the state space, $\mathcal{G}$ is the goal space, $\mathcal{A}$ is the action space, $p(s'|s, a)$ is the transition dynamics, $r(s, a)$ is the reward function, $\gamma \in [0, 1)$ is the discount factor, and $H$ is the horizon.

**Goal-conditioned HRL.** We consider a framework that consists of two hierarchies: a high-level policy $\pi(g|s; \theta_{\texttt{high}})$ and a low-level policy $\pi(a|s, g; \theta_{\texttt{low}})$, where each policy parameterized by neural networks whose parameters are $\theta_{\texttt{high}}$ and $\theta_{\texttt{low}}$, respectively. At each timestep $t$, The high-level policy generates a high-level action, i.e., subgoal $g_t \in \mathcal{G}$, by either sampling from its policy $g_t \sim \pi(g|s_t; \theta_{\texttt{high}})$ when $t \equiv 0 \pmod{k}$, or otherwise using a pre-defined goal transition process $g_t = h(g_{t-1}, s_{t-1}, s_t)$, i.e., $h(g_{t-1}, s_{t-1}, s_t) = g_{t-1} + s_{t-1} - s_t$ for relative subgoal scheme [28, 54], $h(g_{t-1}, s_{t-1}, s_t) = g_{t-1}$ for absolute subgoal scheme.[2] The low-level policy observes the state $s_t$ and goal $g_t$, and performs a low-level atomic action $a_t \sim \pi(a|s_t, g_t; \theta_{\texttt{low}})$. Then, the reward function for the high-level policy is given as the sum of $m$ external rewards from the environment as follows:

$$r^{\texttt{high}}(\tau_{t,m}) = \sum_{i=0}^{m-1} r(s_i, a_i), \tag{1}$$

where $\tau_{t,m} = \{(s_t, g_t, a_t), \cdots, (s_{t+m-1}, g_{t+m-1}, a_{t+m-1})\}$ denotes a trajectory segment of size $m$. The goal of a high-level policy is to maximize the expected sum of $r^{\texttt{high}}$ by providing the low-level policy with an intrinsic reward proportional to the distance in the goal space $\mathcal{G}$. Specifically, in a relative subgoal scheme [28, 54], the reward function for a low-level policy is defined as:

$$r^{\texttt{low}}(s_t, g_t, a_t, s_{t+1}) = -\|g_t, \varphi(s_{t+1} - s_t)\|_2, \tag{2}$$

where $\varphi : \mathcal{S} \to \mathcal{G}$ is a goal mapping function that maps a state to a goal. Instead, if one replaces with an absolute subgoal scheme, the reward function is substituted as follows:

$$r^{\texttt{low}}(s_t, g_t, a_t, s_{t+1}) = -\|g_t, \varphi(s_{t+1})\|_2, \tag{3}$$

**Random network distillation.** One line of exploration algorithms introduce novelty of a state, that is calculated by prediction errors [3, 13, 34, 39], visit-counts [2, 33, 47], or state entropy estimate [12, 18, 27, 40]. One well-known method is Random Network Distillation (RND) [3], which utilizes the prediction error of a neural network as a novelty score. Specifically, let $f$ be a neural network with fixed parameters $\bar{\theta}$ and $\hat{f}$ be a neural network parameterized by $\theta$. RND updates $\theta$ by minimizing the expected mean squared prediction error of $f$, $\mathbb{E}_{s \sim \mathcal{B}} \|\hat{f}(s; \theta) - f(s; \bar{\theta})\|_2$. Then the novelty score of a state $s$ is defined as:

$$n(s) = \|\hat{f}(s; \theta) - f(s; \bar{\theta})\|_2. \tag{4}$$

The novelty score $n(s)$ is likely to be higher for novel states dissimilar to the ones the predictor network $\theta$ has been trained on.

---

[2]In a relative subgoal scheme, the high-level policy gives a subgoal representing how far the low-level policy should move from its current state. Whereas, in an absolute subgoal scheme, the high-level policy provides an *absolute* position where the low-level policy should reach.

**Adjacency network.** Let $d_{\tt st}(s, s')$ be the shortest transition distance from state $s$ to state $s'$, i.e., $d_{\tt st}(s, s')$ is the expected number of steps an optimal agent should take to reach the state $s'$ from $s$. To estimate $d_{\tt st}$, Zhang et al. [54] introduce an *adjacency network* $\psi$ parameterized by $\phi$, that discriminates whether two states are $k$-step adjacent or not. The network learns a mapping from a goal space to an adjacency space by minimizing the following contrastive-like loss:

$$
\begin{aligned}
\mathcal{L}_{\tt adj}(\phi) = \mathbb{E}_{s_i, s_j \in \mathcal{S}}[&l \cdot \max(\|\psi_\phi(g_i) - \psi_\phi(g_j)\|_2 - \varepsilon_k, 0) \\
&+(1 - l) \cdot \max(\varepsilon_k + \delta - \|\psi_\phi(g_i) - \psi_\phi(g_j)\|_2, 0)],
\end{aligned}
\tag{5}
$$

where $\delta > 0$ is a margin between embeddings, $\varepsilon_k$ is a scaling factor, and $l \in \{0, 1\}$ represents the label indicating $k$-step adjacency derived from the $k$-step adjacency matrix $\mathcal{M}$ that stores the adjacency information of the explored states. The equation (5) penalizes adjacent state embeddings ($l = 1$) with large Euclidean distances, while non-adjacent state embeddings ($l = 0$) with small Euclidean distances. Then the shortest transition distance can be estimated as follows:

$$
\widehat{d}_{\tt st}(s, s'; \phi) = \frac{k}{\varepsilon_k} \|\psi_\phi(g_1), \psi_\phi(g_2)\|_2 \approx d_{\tt st}(s, s').
\tag{6}
$$

## 4 Hierarchical reinforcement learning guided by landmarks (HIGL)

In this section, we propose HIGL: **HI**erarchical reinforcement learning **G**uided by **L**andmarks, a novel framework for training a high-level policy with reduced action space guided by landmarks. We describe HIGL with three parts sequentially: (1) landmark sampling in Section 4.1, (2) landmark selection in Section 4.2, and (3) training in Section 4.3. We provide an illustration and an overall description of our framework in Figure 1 and Algorithm **??**, respectively.

### 4.1 Landmark sampling

To effectively guide the subgoal generation of a high-level policy, it is important to construct a set of landmarks that covers a wide area of state space and contains novel states promising to explore. To this end, we consider two criteria for landmark selection: (1) the coverage of the entire visited state space and (2) the novelty of a state.

**Coverage-based sampling.** We propose to sample landmarks that cover a wide range of visited states from a replay buffer $\mathcal{B}$. To this end, we utilize Farthest Point Sampling (FPS) [49], which samples a *pool* of states from $\mathcal{B}$ and chooses states which are as far as possible from each other in the pool. Specifically, we sample a set of *coverage-based* landmarks $L^{\tt cov} = \{l_i^{\tt cov}\}_{i=1}^{M_{\tt cov}}$ by applying FPS where the distance between two states $s$ and $s'$ is measured in the goal space as $\|\varphi(s) - \varphi(s')\|_2$, following Huang et al. [14]. We note that FPS implicitly samples states at the frontier of visited state space, which implies that coverage-based landmarks implicitly mean promising states to explore as well (see Figure 6 for supporting experimental results).

**Novelty-based sampling.** To explicitly sample novel landmarks, we propose to store the novel states encountered during the environment interaction and utilize them as landmarks. To this end, we introduce a priority queue $\mathcal{Q}$ of a fixed size $K$ where the priority of each element (state) is defined as the novelty of a state $n(s)$ in (4); the queue stores a state $s$ in $\mathcal{Q}$ with a priority of $n(s)$. One important thing here is that the novelty of a state $s$ decreases as the exploration proceeds, so the priority of stored states in $\mathcal{Q}$ should be constantly updated. For this reason, we propose a similarity-based update scheme that discards previously-stored samples that are similar to the newly encountered state. Specifically, when we encounter a state $s$, we measure the similarity of $s$ between all stored states $s' \in \mathcal{Q}$ and discard similar states, i.e., $\{s' \in \mathcal{Q} : \|\varphi(s) - \varphi(s')\|_2 < \lambda\}$, where $\lambda$ is a similarity threshold. Then we store a state $s$ in $\mathcal{Q}$, and sample a set of *novelty-based* landmarks $L^{\tt nov} = \{l_i^{\tt nov}\}_{i=1}^{M_{\tt nov}}$ from $\mathcal{Q}$.

### 4.2 Landmark selection

Since all the landmarks in $L = L^{\tt cov} \cup L^{\tt nov}$ are not equally valuable to reach a goal $g$ from a current state $s$, i.e., some landmarks may be irrelevant or even impeditive to arrive at the goal, we propose a landmark selection scheme to select the most *urgent* landmark among the sampled landmarks. To

this end, we introduce a two-stage scheme: (a) we first build a graph of landmarks, and (b) we run the shortest path planning to a goal in the graph.

**Building a graph.** For landmark selection, we build a graph whose nodes consist of a current state $s_t$, a (final) goal $g$, and landmarks $L$. First, we connect all the nodes and assign the weight of each edge with a distance between two nodes, where distance is estimated by a low-level (goal-conditioned) value function $V(s, g)$, i.e., $-V(s_1, \varphi(s_2 - s_1))$ for state $s_1, s_2$ in a relative subgoal scheme, following prior works [8, 14, 32]. As the distance estimation via value function is locally accurate but unreliable for far states, we disconnect two nodes when the weight of the corresponding edge is larger than a preset threshold $\gamma_{\mathtt{dist}}$ following Huang et al. [14].

**Planning.** After building a graph, we run the shortest path planning algorithm to select the most urgent state to visit, $l_t^{\mathtt{sel}}$, from a current state $s_t$ to a goal $g$. Since the general value iteration for RL problems is exactly the shortest path algorithm on the graph, we utilize the value iteration as the shortest path planning, following the prior work of Huang et al. [14]. By selecting the very first landmark in the shortest path to the goal, HIGL can focus on the most urgent landmark, ignoring landmarks that may be irrelevant to reach the goal from the current state.

### 4.3 Training high-level policy guided by landmark

Using the selected landmark, HIGL trains high-level policy to generate subgoals that satisfy both desired properties: (1) *reachable* from the current state and (2) toward promising states to explore. Since the *raw* selected landmark may be placed far from the current state, using the raw landmark would be suboptimal; it is likely not to satisfy the property (1). To acquire both properties, we introduce *pseudo*-landmark, which is located near the current state but also directed toward the selected landmark (promising state to explore) in the goal space. To be specific, we make pseudo-landmark be placed between the selected landmark and the current state in the goal space as follows:

$$g_t^{\mathtt{pseudo}} := g_t^{\mathtt{cur}} + \delta_{\mathtt{pseudo}} \cdot \frac{g_t^{\mathtt{sel}} - g_t^{\mathtt{cur}}}{||g_t^{\mathtt{sel}} - g_t^{\mathtt{cur}}||_2}, \tag{7}$$

where $\delta_{\mathtt{pseudo}}$ is the shift magnitude, and $g_t^{\mathtt{sel}} = \varphi(l_t^{\mathtt{sel}})$, $g_t^{\mathtt{cur}} = \varphi(s_t)$ are the corresponding points for the selected landmark $l_t^{\mathtt{sel}}$ and the current state $s_t$ in the "goal" space, respectively.

Then, we encourage high-level policy to generate a subgoal adjacent to the pseudo-landmark. To discriminate the adjacency, we employ the adjacency network proposed in Zhang et al. [54]. Instead of a strict adjacency constraint, which may cause instability in training, we "encourage" high-level policy to generate a subgoal near pseudo-landmark via *landmark loss*, motivated by the prior work [54]. The landmark loss is calculated using the adjacency network as follows:

$$\mathcal{L}_{\mathtt{landmark}}(\theta_{\mathtt{high}}) = \max(||\psi_\phi(g_t^{\mathtt{pseudo}}) - \psi_\phi(g_t)||_2 - \varepsilon_k, 0), \tag{8}$$

where $g_t \sim \pi(g|s_t; \theta_{\mathtt{high}})$ is a generated subgoal by the high-level policy, $k$ is adjacency degree (how far we admit as adjacency), and $\varepsilon_k$ is a corresponding scaling factor. Then, we train high-level policy by incorporating $\mathcal{L}_{\mathtt{landmark}}$ into the goal-conditioned HRL framework:

$$\mathcal{L}_{\mathtt{high}}(\theta_{\mathtt{high}}) = -\mathbb{E}_{\theta_{\mathtt{high}}} \sum_{t=0}^{T-1} (\gamma^t r^{\mathtt{high}}(\tau_{t,m}) - \eta \cdot \mathcal{L}_{\mathtt{landmark}}), \tag{9}$$

where $\eta$ is the balancing coefficient. In practice, we plug $\mathcal{L}_{\mathtt{landmark}}$ as an extra loss term into the original policy loss term of a specific high-level RL algorithm, e.g., TD error for temporal-difference learning methods. We remark that the low-level policy is trained as usual without any modification.

## 5 Experiments

In this section, we designed our experiments to answer the following questions:

- How does HIGL compare to the state-of-the-art HRL method [54] across various long-horizon continuous control tasks (see Figure 3)?
- How do the coverage and novelty for landmark sampling improve the performance, respectively (see Figure 4a, 4b)?

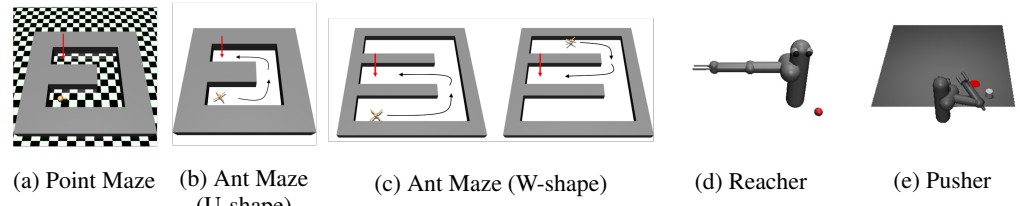

(a) Point Maze    (b) Ant Maze    (c) Ant Maze (W-shape)    (d) Reacher    (e) Pusher
(U-shape)

Figure 2: Environments used in our experiments. In maze tasks, the red arrow indicates the goal in each task, and the black line represents the desired trajectory from the current state to the goal. In (a) Point Maze and (b) Ant Maze (U-shape), an agent is born at the bottom-left corner at the start of the episode. In (c) Ant Maze (W-shape), an agent is born at a random point in the maze except for the goal point. In (d) Reacher and (e) Pusher, a robotic arm aims to make its end-effector and (puck-shaped) gray object reach the target position, which is marked as a red ball, respectively.

- How does the pseudo-landmark affect the performance instead of using the raw selected landmarks (see Figure 4c)?
- How do the hyperparameters: (1) the number of landmarks $M$, (2) the shift magnitude $\delta_{\texttt{pseudo}}$, and (3) the adjacency degree $k$ affect the performance (see Figure 5a, 5b, 5c)?

## 5.1 Experimental setup

**Environments.** We conduct our experiments on a set of challenging long-horizon continuous control tasks based on MuJoCo simulator [48]. Specifically, we consider the following environments to evaluate our framework (see Figure 2 for the visualization of environments).

- **Point Maze** [7]: A simulated ball starts at the bottom left corner in a "⊃"-shaped maze and aims to reach the top left corner.
- **Ant Maze (U-shape)** [7]: A simulated ant starts at the bottom left corner in a "⊃"-shaped maze and aims to reach the top left corner.
- **Ant Maze (W-shape)** [54]: A simulated ant starts from a random position in the "∃"-shaped maze and aims to reach the target position located at the middle left corner.
- **Reacher** [4]: A robotic arm aims to make its end-effector reach the target position.
- **Pusher** [4]: A robotic arm aims to make a (puck-shaped) object in a plane reach a goal position by pushing the object.
- **Stochastic Ant Maze (U-shape)** [54]: Gaussian noise with standard deviation $\sigma$ (i.e., 0.05) is added to the $(x, y)$ position of the ant robot at every step.

Moreover, we evaluate HIGL with two different reward shapings *dense* and *sparse*. In the dense reward shaping, the reward is the negative L2 distance from the current state to the target position (final goal) in the goal space. In the sparse setting, the reward is 0 if the distance to the target position is lower than a pre-set threshold, otherwise -1. In maze environments, we use a pre-defined 2-dimensional goal space that represents the $(x, y)$ position of the agent following prior works [28, 54]. In Reacher, we use 3-dimensional goal space that represents the $(x, y, z)$ position of the end-effector. In Pusher, we use 6-dimensional space, which additionally includes the 3D position of the (puck-shaped) object. We employ a relative subgoal scheme for Maze tasks and an absolute one for Reacher and Pusher. We provide more environmental details in the supplementary material.

**Implementation.** We use TD3 algorithm [10] as the underlying algorithm for training both high-level policy and low-level policy for all considered methods. For the number of coverage-based landmarks $M_{\texttt{cov}}$ and the number of novelty-based landmarks $M_{\texttt{cov}}$, we use $M_{\texttt{cov}} = 20$ and $M_{\texttt{nov}} = 20$ in all the environments except Ant Maze (W-shape). We use $M_{\texttt{cov}} = 40$ and $M_{\texttt{nov}} = 40$ in the more complex Ant Maze (W-shape) environment. In order to avoid the instability in training due to the noisy pseudo-landmark in the early phase of training, we use $\delta_{\texttt{pseudo}} = 0$ for the initial 60K timesteps, i.e., $k$-step adjacent region to the "current state" instead of "pseudo-landmark." We find that this stabilizes the training by avoiding inaccurate planning at the early phase; under-trained value function causes inaccurate distance estimation at landmark selection. All of the experiments were processed using a single GPU (NVIDIA TITAN Xp) and 8 CPU cores (Intel Xeon E5-2630 v4). We evaluate five test episodes without an exploration factor for every $5000^{\text{th}}$ time step. We provide further implementation details used for our experiments in the supplementary material.

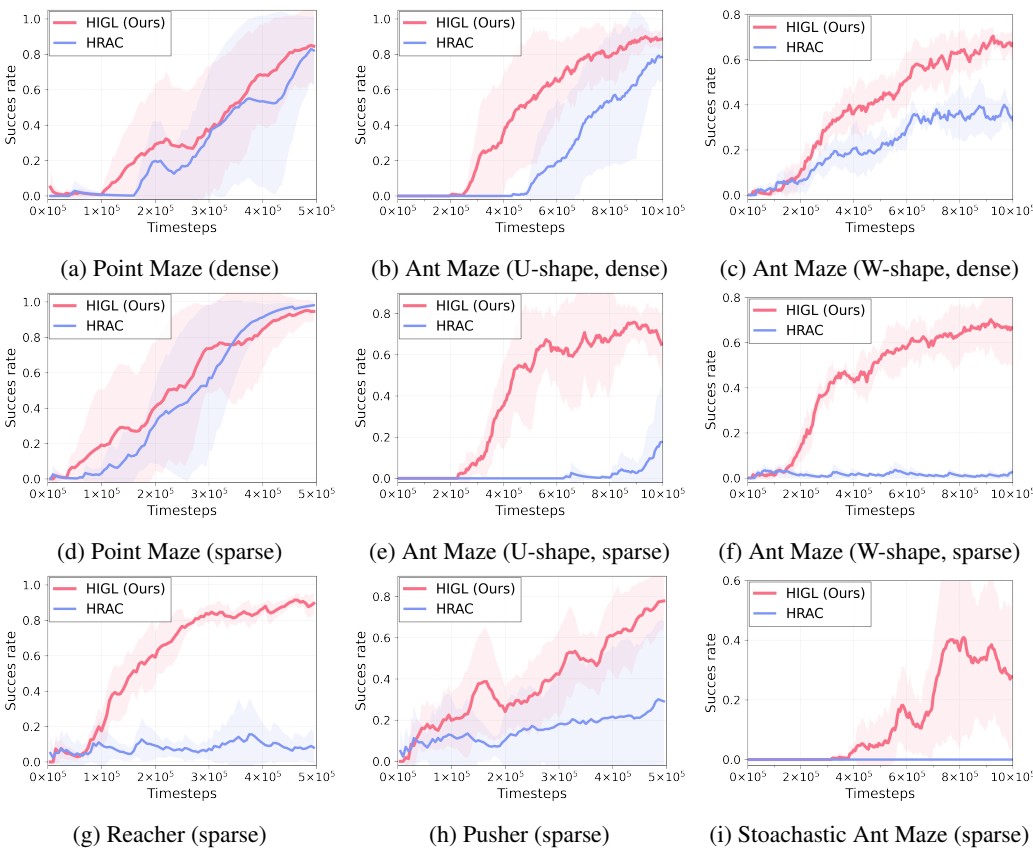

Figure 3: The average success rate in various continuous control tasks of HIGL and HRAC [54]. We observe that HIGL consistently outperforms HRAC, especially in more complex and long-horizon environments. The solid lines and shaded regions represent mean and standard deviation, respectively, across four runs. All curves are smoothed equally for visual clarity.

## 5.2 Comparative evaluation

We compare HIGL to the prior state-of-the-art method HRAC, which encourages a high-level policy to generate a subgoal within the $k$-step adjacent region of the current state. As shown in Figure 3, HIGL is very effective in hard-exploration tasks (i.e., Ant Maze (U-shape and W-shape)) thanks to its efficient exploration guided by landmarks. To be specific, instead of treating all the adjacent states equally (as HRAC did), HIGL considers both reachability and the novelty of a state. HIGL recognizes promising directions to explore via planning and trains high-level policy to generate a subgoal toward the direction. We understand that such differences in our mechanism made a large gain over HRAC. In particular, HIGL achieves a success rate of 65.1%, whereas HRAC performs about 17.6% at timesteps $10 \times 10^5$ in Ant Maze (U-shape, sparse) task. We emphasize that HIGL is more sample-efficient when the task is much difficult; HIGL shows a larger margin in performance in (1) Ant Maze (U-shape) than Point Maze, and (2) sparse reward setting than dense reward setting.

Moreover, We find that HIGL also outperforms HRAC in stochastic environments, as shown in Figure 3i. We remark that HIGL is applicable to stochastic environments without any modification since our algorithmic components (including the novelty priority queue and a landmark-graph) are built on visited states, regardless of transition dynamics.

## 5.3 Ablation studies

We conduct ablation studies on HIGL to investigate the effect of (1) coverage-based and novelty-based sampling in Figure 4a, 4b, (2) pseudo-landmarks (compared to raw selected landmarks) in Figure 4c, and (3) hyperparameters (i.e., number of landmarks $M$, shift magnitude $\delta_{\texttt{pseudo}}$, and adjacency

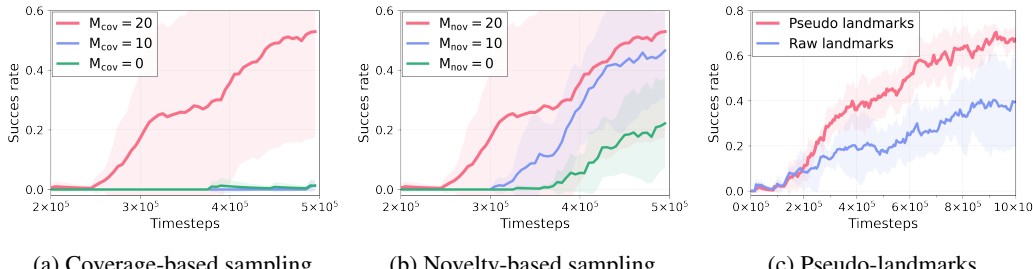

| (a) Coverage-based sampling | (b) Novelty-based sampling | (c) Pseudo-landmarks |

Figure 4: Ablation studies on our algorithmic components: (a) coverage-based sampling, (b) novelty-based sampling, and (c) pseudo-landmarks. We measure the performance of HIGL by varying the number of (a) coverage-based and (b) novelty-based landmarks in Ant Maze (U-shape, dense). For (c), we compare HIGL with pseudo-landmarks and raw landmarks in Ant Maze (W-shape, dense).

degree $k$) in Figure 5. For all the ablation studies except for the pseudo-landmarks, we use Ant Maze (U-shape, dense). For experiments on the pseudo-landmarks, we employ Ant Maze (W-shape, dense), which has a larger maze size (i.e., W-shape has a size of $20 \times 20$, while U-shape has $12 \times 12$) because the effectiveness of the pseudo-landmarks is more remarkable in such a large map.

**Coverage-based sampling.** We evaluate HIGL with varying numbers of samples from coverage-based sampling in Figure 4a. We evaluate HIGL with $M_{\text{nov}} = 20$ and varying $M_{\text{cov}} \in \{0, 10, 20\}$. We observe that using coverage-based landmarks affects the performance of HIGL, indeed. This is because coverage-based landmarks play an important role as waypoints toward novel states or even as promising states themselves; coverage-based sampling implicitly samples states at the frontier of visited state space (See Figure 6 for supporting qualitative analysis).

**Novelty-based sampling.** Analogously, we evaluate HIGL with varying numbers of samples from novelty-based sampling in Figure 4b. Specifically, we report the performance of HIGL with $M_{\text{cov}} = 20$ and varying $M_{\text{nov}} \in \{0, 10, 20\}$. We observe that utilizing our proposed novelty-based sampling improves the performance as well. For example, we emphasize that HIGL with $M_{\text{nov}} = 10$ novelty-based landmarks significantly improves over $M_{\text{nov}} = 0$, which corresponds to HIGL with only coverage-based sampling. This demonstrates the importance of considering the novelty of each state is crucial for efficient exploration.

**Pseudo-landmarks.** To recognize the effectiveness of pseudo-landmarks, compared to raw selected landmarks, we conduct ablative experiments in Ant Maze (W-shape, dense). As shown in Figure 4c, we observe that using pseudo-landmarks achieves better performance than using raw selected landmarks. This is because pseudo-landmarks have both desired properties: (1) "reachable" from the current state and (2) toward promising states to explore, while the "raw" selected landmarks may only have the latter property. When a selected landmark is placed too far from the current state, using the selected one would be suboptimal because it would make high-level policy generate unreachable subgoals from the current state; providing such subgoal makes a faint reward signal for low-level policy. Instead, pseudo-landmarks can effectively guide high-level policy with both desired properties, so it accelerates training even in such a large environment like Ant Maze (W-shape), where selected landmarks are more likely to be located far from the current state.

**Hyperparameters.** We conduct experiments to verify the effectiveness of hyperparameters, (1) number of landmarks $M = M_{\text{cov}} + M_{\text{nov}}$, (2) shift magnitude $\delta_{\text{pseudo}}$, and (3) adjacency degree $k$.

- **Number of landmarks $M$.** To demonstrate the effectiveness of the number of landmarks, we conduct experiments using the different number of landmarks $M = M_{\text{cov}} + M_{\text{nov}}$ in Figure 5a. We sample the same number of landmarks for each criterion, i.e., $M_{\text{cov}} = M_{\text{nov}}$. The results show that the performance of HIGL is improved with the increased number of landmarks since it is more capable of containing more information about an environment. In addition, one can understand that increasing the number makes planning in the landmark-graph more reliable; we remark that distance estimation via (low-level) value function is more accurate in the local area.

- **Shift magnitude $\delta_{\text{pseudo}}$.** In Figure 5b, we conduct experiments with varying values of $\delta_{\text{pseudo}}$, which determines the location of pseudo-landmark. The results demonstrate the location of the

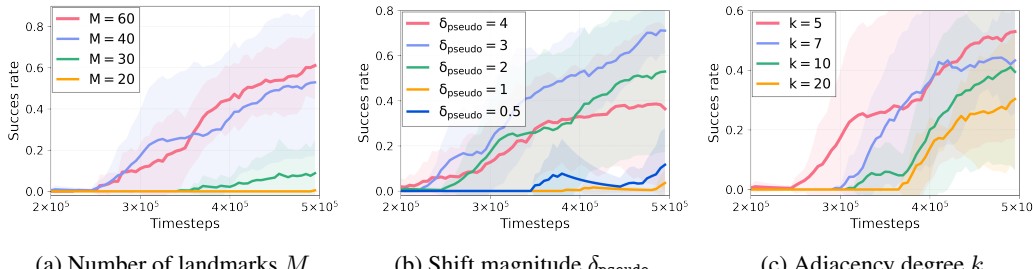

(a) Number of landmarks $M$      (b) Shift magnitude $\delta_{\text{pseudo}}$      (c) Adjacency degree $k$

Figure 5: Performance of HIGL on Ant Maze (U-shape, dense) environment with varying (a) the number of landmarks $M$, (b) shift magnitude $\delta_{\text{pseudo}}$, and (c) adjacency degree $k$.

pseudo landmark can affect the performance. If $\delta_{\text{pseudo}}$ is small, the high-level policy tends to be trained to generate a subgoal near the current state rather than the selected landmark; this may cause the high-level policy to not fully enjoy the benefits of efficient exploration guided by the selected landmark. On the other hand, if $\delta_{\text{pseudo}}$ is too large, the high-level policy is promoted to generate a subgoal that is unreachable from the current state, which leads to performance degradation, i.e., $\delta_{\text{pseudo}} = 4$ in Figure 5b.

- **Adjacency degree $k$.** In Figure 5c, we investigate the effectiveness of the adjacency degree $k$, which determines the size of the region where high-level policy is encouraged to generate a subgoal. One can observe that adjusting adjacency degree does not influence critically to achieve superior performance over HRAC. However, we understand that setting the degree too large is not appropriate because it is allowed for high-level policy to generate a subgoal that is quite far from the pseudo landmark; the generated subgoal may be located unreachable region, which gives a faint signal to a low-level policy, i.e., $k = 20$ in Figure 5c.

### 5.4 Qualitative analysis on landmark sampling

In Figure 6, we qualitatively analyze how our landmarks sampling method works in the Ant Maze (U-shape, dense) task. We sample 20 coverage-based landmarks (blue dots) and 20 novelty-based landmarks (red dots); then, we visualize them in the goal space (i.e., 2D space). One can find that coverage-based landmarks are dispersed across visited space, and novelty-based ones are concentrated to the frontier of the visited space. In particular, at the early phase of training, the coverage-based landmarks are scattered in the bottom-left region of the maze since an agent is born at the bottom-left corner in the beginning of an episode. As training proceeds, the agent visits wider regions, so coverage based-landmarks are more scattered across the map. Remarkably, the novelty-based landmarks are concentrated at the frontier of the visited space throughout the training phase.

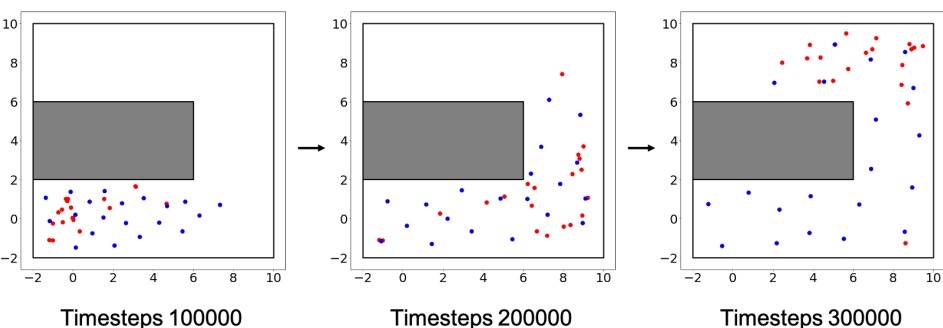

Figure 6: Qualitative analysis on the landmark sampling scheme in HIGL. The blue dots and the red dots denote coverage-based landmarks and novelty-based landmarks in the goal space (i.e., 2D space), respectively. One can observe that the coverage-based landmarks are dispersed across the visited state space, and novelty-based landmarks are concentrated at the frontier of the visited space, which is likely to be novel. The analysis is conducted on the Ant Maze (U-shape, dense).

# 6 Discussion and conclusion

We present HIGL, a new framework for training a high-level policy with reduced action space guided by landmarks. Our main idea is (a) sampling landmarks that are informative for exploration and (b) training the high-level policy to generate a subgoal toward a selected landmark. Experiments show that HIGL outperforms the prior state-of-the-art method thanks to efficient exploration by landmarks. We believe that our framework would guide a new interesting direction in the HRL: high-level action space reduction into the promising region to explore.

One interesting future work of HIGL would be an application to environments with high-dimensional state spaces (i.e., image-based environments). In principle, HIGL is applicable to such environments, but one potential issue is that the required number of landmarks would be increased. The increased number of landmarks can lead to spending more time in planning over a landmark-graph. To alleviate this issue, one can build the priority queue and the landmark-graph in "goal space" instead of "state space"; goal space typically has a lower dimension. The reason why one can build them in "goal space" comes from the fact that HIGL eventually utilizes landmarks in "goal space" rather than "state space," as equation 8 shows. We expect that HIGL combined with subgoal representation learning (which learns state to goal mapping function) would be successful since it has shown promising performance on environments with high-dimensional state spaces [21, 29, 46].

**Limitation.** While our experiments demonstrate that HIGL is effective for solving complex control tasks, we only consider the setup where the final goal of a task is given. An interesting future direction is to develop a landmark selection scheme that works without the final goal, i.e., selecting the very first landmark in the shortest path to a *pseudo* final goal, (i.e., hindsight goal).

Also, one may point out that the cost consumed for planning may cause a scalability issue, as we perform the shortest path planning at every training step. To be specific, for 1M training timesteps, HIGL takes 13 hours, and HRAC takes 6 hours using a single GPU (NVIDIA TITAN Xp) and 8 CPU cores (Intel Xeon E5-2630 v4). However, given that (i) our method does not utilize planning at deployment time where the execution response time is important, and (ii) environment interaction for sample collection is often dangerous and expensive, we believe that incurring such costs to improve the response time and the sample-efficiency of the algorithm is a reasonable and appropriate direction.

**Potential negative impacts.** This work would promote the research in the field of HRL and has potential real-world applications such as robotics. However, there could be potential negative consequences of developing an algorithm for autonomous agents. For example, if a malicious user specifies a reward function that corresponds to harmful behavior for a society, an RL agent would just learn such behaviors without considering the expected results. Specifically, developing an HRL agent for solving a complex and long-term task would facilitate the development of the real-world deployment of malicious robots, which could perform a long-horizon operation in an autonomous way without the direction of a human. For this reason, in addition to developing an HRL and RL algorithms for improving the sample efficiency and performance, it is important to devise a method that could consider the consequence of its own behaviors to a society.

## Acknowledgments and Disclosure of Funding

We thank Kimin Lee and anonymous reviewers for providing helpful feedback and suggestions in improving our paper. This work was supported by Institute of Information & communications Technology Planning & Evaluation (IITP) grant funded by the Korea government (MSIT) (No.2019-0-00075, Artificial Intelligence Graduate School Program (KAIST)) and the Engineering Research Center Program through the National Research Foundation of Korea (NRF) funded by the Korean Government MSIT (NRF-2018R1A5A1059921).

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
