# A  Algorithm table

We provide an algorithm table that represents HIGL in Algorithm 1.

---

**Algorithm 1** Hierarchical reinforcement learning guided by landmarks (HIGL)

---

**Input:** Goal transition function $h$, state-goal mapping function $\varphi$, high-level action frequency $m$, the number of training episode $N$, adjacency learning frequency $C$, replay buffer $\mathcal{B}$, training batch size $B$ and the number of landmarks $M_{\texttt{cov}}$, $M_{\texttt{nov}}$
Initialize the parameters of high-level policy $\theta_{\texttt{high}}$, low-level policy $\theta_{\texttt{low}}$, adjacency network $\phi$, RND networks $\theta, \bar{\theta}$
Initialize empty adjacency matrix $\mathcal{M}$
Initialize priority queue $\mathcal{Q}$
**for** $n = 1, \ldots, N$ **do**
    Reset the environment and sample the initial state $s_0$.
    $t = 0$.
    **repeat**
        **if** $t \equiv 0 \pmod{m}$ **then**
            Sample subgoal $g_t \sim \pi(g|s_t; \theta_{\texttt{high}})$.
        **else**
            Perform subgoal transition $g_t = h(g_{t-1}, s_{t-1}, s_t)$
        **end if**
        Collect a transition $(s_t, a_t, s_{t+1}, r_t)$ using low-level policy $\theta_{\texttt{low}}$.
        Calculate novelty of the state $s_t$ using RND networks $\theta, \bar{\theta}$ and update the priority queue $\mathcal{Q}$.
        Sample episode end signal $done$.
        $t = t + 1$
    **until** $done$ is $true$
    Store the sampled trajectory in $\mathcal{B}$.
    **for** $j = 1, \ldots, B$ **do**
        Sample a state and a corresponding goal from $\mathcal{B}$.
        Sample $M_{\texttt{cov}}$ landmarks from $\mathcal{B}$ and $M_{\texttt{nov}}$ landmarks from $\mathcal{Q}$, and merge them.
        Build a graph with the sampled landmarks, a state and a goal.
        Select a landmark in the graph. (i.e., the very first landmark in the shortest path to a goal.)
        Train high-level policy $\theta_{\texttt{high}}$ according to equation 9.
        Train low-level policy $\theta_{\texttt{low}}$.
        Train RND network $\theta$.
    **end for**
    **if** $n \equiv 0 \pmod{C}$ **then**
        Update the adjacency matrix $\mathcal{M}$ using trajectories in $\mathcal{B}$.
        Train $\phi$ using $\mathcal{M}$ by minimizing equation 5.
    **end if**
**end for**

---

# B Environment details

## B.1 Point Maze

A simulated ball (point mass) starts at the bottom left corner in a "⊃"-shaped maze and aims to reach the top left corner. In detail, the environment has a size of $12 \times 12$, with a continuous state space including the current position and velocity, the current timestep $t$, and the target location. The dimension of actions is two; one action determines a rotation on the pivot of the point mass, and the other action determines a push or pull on the point mass in the direction of the pivot. At training time, a target position is sampled uniformly at random from $g_x \sim [-2, 10], g_y \sim [-2, 10]$. At evaluation time, we evaluate the agent only its ability to reach $(0, 8)$. We define a 'success' as being within an L2 distance of 2.5 from the target. Each episode terminates at 500 steps.

## B.2 Ant Maze (U-shape)

This environment is equivalent to the Point Maze except for the substitution of the point mass with a simulated ant. Its actions correspond to torques applied to joints. All the other detail, such as the goal generation scheme and definition of "success", are the same as the Point Maze.

## B.3 Ant Maze (W-shape)

This environment has a "∃"-shaped maze whose size is $20 \times 20$, with the same state and action spaces as the Ant Maze (U-shape) task. The target position $(g_x, g_y)$ is set at the position $(2, 9)$ in the center corridor at both training and evaluation time. At the beginning of each episode, the agent is randomly placed in the maze except at the goal position. We define a "success" as being within an L2 distance of 1.0 from the target. Each episode is terminated if the agent reaches the goal or after 500 steps.

## B.4 Reacher & Pusher

Each episode terminates at 100 steps. We define a "success" as being within an L2 distance of 0.25 from the target. Reacher has a continuous state space of which dimension is 17, including the positions, angles, velocities of the robot arm, and the goal position. Pusher additionally includes the 3D position of a puck-shaped object, so it has 20-dimensional state space. The environments have 7-dimensional action space, of which range is $[-20, 20]$ in Reacher and $[-2, 2]$ in Pusher. In addition, there exists an action penalty in Reacher and Pusher; the penalty is the squared L2 distance of the action and is multiplied by a coefficient of 0.0001 in Reacher and 0.001 in Pusher. Then, the penalty is deducted from the reward.

# C Implementation details

## C.1 Network structure

For the hierarchical policy network, we employ the same architecture as HRAC [1], where both the high-level and the low-level use TD3 [2] algorithm for training. Each actor and critic network for both high-level and low-level consists of 3 fully connected layers with ReLU nonlinearities. The size of each hidden layer is $(300, 300)$. The output of the high-level and low-level actor is activated using the `tanh` function and is scaled to the range of corresponding action space.

For the adjacency network, we employ the sample architecture as HRAC [1], where the network consists of 4 fully connected layers with ReLU nonlinearities. The size of each hidden layer is $(128, 128)$. The dimension of the output embedding is 32.

For RND, the network consists of 3 fully connected layers with ReLU nonlinearities. The size of the hidden layers of the RND network is $(300, 300)$. The dimension of the output embedding is 128.

We use Adam optimizer [3] for all networks.

## C.2 Training parameters

We list hyperparameters for hierarchical policy, adjacency network, and RND network used across all environments in Table 1 and 2. Hyperparameters that differ across the environments are in Table 3.

Table 1: Hyperparameters for hierarchical policy across all environments.

| Hyperparameter | Value | Value |
|---|---|---|
| High-level TD3 | Low-level TD3 | |
| Actor learning rate | 0.0001 | 0.0001 |
| Critic learning rate | 0.001 | 0.001 |
| Replay buffer size | 200000 | 200000 |
| Batch size | 128 | 128 |
| Soft update rate | 0.005 | 0.005 |
| Policy update frequency | 1 | 1 |
| $\gamma$ | 0.99 | 0.95 |
| Reward scaling | 0.1 | 1.0 |
| Landmark loss coefficient $\eta$ | 20 | |

Table 2: Hyperparameters for adjacency network and RND network across all environments.

| Hyperparameter | Value |
|---|---|
| Adjacency network | |
| Learning rate | 0.0002 |
| Batch size | 64 |
| $\varepsilon_k$ | 1.0 |
| Training frequency (steps) | 50000 |
| Training epochs | 25 |
| RND network | |
| Learning rate | 0.001 |
| Batch size | 128 |

Table 3: Hyperparameters that differ across the environments.

| Hyperparameter | Point Maze | Ant Maze (U-shape) | Ant Maze (W-shape) | Reacher & Pusher |
|---|---|---|---|---|
| **High-level TD3** | | | | |
| High-level action frequency $m$ | 10 | 10 | 10 | 5 |
| Exploration strategy | Gaussian ($\sigma = 1.0$) | Gaussian ($\sigma = 1.0$) | Gaussian ($\sigma = 1.0$) | Gaussian ($\sigma = 0.2$) |
| $M_{\text{cov}}, M_{\text{nov}}$ | 20 | 20 | 60 | 20 |
| Similarity threshold $\lambda$ | 0.2 | 0.2 | 0.2 | 0.02 |
| $\gamma_{\text{dist}}$ | 38.0 | 38.0 | 38.0 | 15.0 |
| Shift magnitude $\delta_{\text{pseudo}}$ | 0.5 | 2.0 | 2.0 | 1.0 |
| Adjacency degree $k$ | 7 | 5 | 5 | 5 |
| **Low-level TD3** | | | | |
| Exploration strategy | Gaussian ($\sigma = 1.0$) | Gaussian ($\sigma = 1.0$) | Gaussian ($\sigma = 1.0$) | Gaussian ($\sigma = 0.1$) |
| **Adjacency network** | | | | |
| $\delta$ | 0.2 | 0.2 | 0.2 | 0.02 |

## D  Additional experiments

Additionally, we provide ablation studies conducted on Ant Maze (U-shape, **sparse**) instead of Ant Maze (U-shape, **dense**). We investigate the effect of (1) coverage-based sampling, (2) novelty-based sampling, (3) the number of landmarks $M = M_{\text{cov}} + M_{\text{nov}}$, (4) shift magnitude $\delta_{\text{pseudo}}$, and (5) adjacency degree $k$ in Figure 1. Overall, one can observe that tendency from Ant Maze (U-shape, **sparse**) and Ant Maze (U-shape, **dense**) are similar.

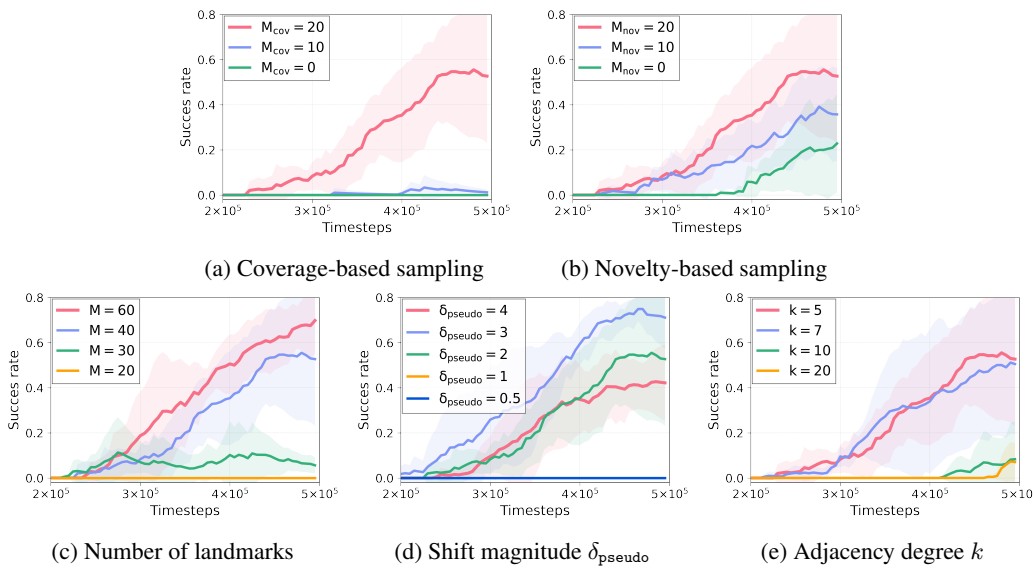

(a) Coverage-based sampling     (b) Novelty-based sampling

(c) Number of landmarks     (d) Shift magnitude $\delta_{\text{pseudo}}$     (e) Adjacency degree $k$

Figure 1: Performance of HIGL on Ant Maze (U-shape, sparse) environment with varying number of (a) coverage-based landmarks $M_{\text{cov}}$ and (b) novelty-based landmarks, $M_{\text{nov}}$, (c) the total number of landmarks $M = M_{\text{cov}} + M_{\text{nov}}$, (d) shift magnitude $\delta_{\text{pseudo}}$, and (e) adjacency degree $k$.

**Discarding design in the priority queue $\mathcal{Q}$.** One can choose another design choice of discarding old states in the novelty priority queue rather than the original design based on the L2-norm in goal-space; for example, one can take discarding design based on the shortest transition distance, i.e., $\hat{d}_{\mathtt{st}}(s, s') < \lambda$. To verify the effectiveness of the discarding design choices, we empirically compare the original discarding design to the alternative design based on the shortest transition distance estimated by the adjacency network. As shown in Figure 2, even though our original design choice shows slightly better performance, both of them outperform the baseline, HRAC.

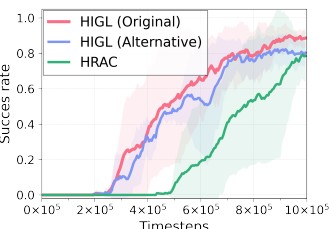

Figure 2: Discarding design

**Automatic shift magnitude.** One can set shift magnitude $\delta_{\mathtt{pseudo}}$ in a systematic manner instead of a pre-set value. Here, one important point is to set "balanced" shift magnitude; too large magnitude would make pseudo-landmarks unreachable, whereas too small magnitude makes no explorative benefits. To this end, for example, one can set $\delta_{\mathtt{pseudo}} = \mathbb{E}\|g_t^{\mathtt{sel}} - g_t^{\mathtt{cur}}\|_2$. Namely, it is the average of the distance between selected landmarks and the current state in the goal space. As shown in Figure 3, using automatic shift magnitude surpasses HRAC. It would be an interesting research direction to improve the automatic manner of setting shift magnitude in the future.

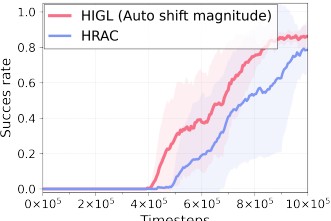

Figure 3: Automatic $\delta_{\mathtt{pseudo}}$

**Larger maze with extended timestep.** We evaluate HIGL on a larger Ant Maze (U-shape) whose size is $24 \times 24$ rather than $12 \times 12$ with extended timesteps of $50 \times 10^5$ in Figure 4. One can observe that HIGL shows highly sample-efficient over the prior state-of-the-art method, HRAC, while both have similar asymptotic performance. We expect that HIGL would be much beneficial in tasks where interaction for sample collection is dangerous and expensive because HIGL could achieve near-asymptotic performance with a relatively small number of samples. We increase the number of landmarks to $M_{\mathtt{cov}} = 40$ and $M_{\mathtt{nov}} = 40$ since the maze is larger than before.

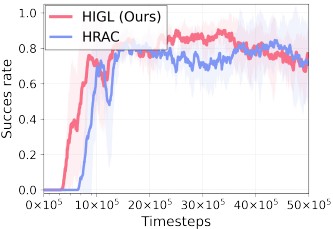

Figure 4: Larger maze