# OpenReview forum: "Landmark-Guided Subgoal Generation in Hierarchical Reinforcement Learning"
_NeurIPS.cc/2021/Conference — NeurIPS 2021 Poster_

### Official Review · Reviewer_RGnx · 2021-07-09

**Rating:** 7
**Confidence:** 4

**Summary:**

Paper introduces HIGL, a hierarchical RL framework based on landmarks. The main idea is to limit the higher level action space (which is often taken to be the full state space), by restricting it to specific landmarks. These landmarks are defined bases on two criteria: coverage of the visited state space, and novelty of the visited states. These landmarks are then integrated in an HRL framework. Experiments on three U-maze like environments show that their method performs well.

**Limitations And Societal Impact:**

Accurately addressed.

**Main Review:**

Strong:
* I like Figure 1: really strong illustration of the overall method, very clear as well.
* The approach to sample landmarks is interesting, especially the combination of both methods (coverage and novelty). Although the authors mention that the coverage method will also mostly sample the frontier (which the novelty method should already do, the coverage method should also cover the interior). Nevertheless, this is a good idea, and Figure 6 gives a really nice illustration that it works.
* There are good results. I like Fig 4b, since it gives a nice illustration of what the interpolation towards the landmark does, and why shooting too far is suboptimal as well. Nevertheless, it does seem beneficial to shoot a bit beyond the landmark (\delta > 1).

Weak/comments:
* The preliminaries on adjancency networks and random network distillation (L92-105) go too fast, and need some additional explanation for reader who have not read those papers. For example, I think you need some explanation on the contrastive loss, and in Eq. 5, where did \bar{\theta} go (is it in f(s)?).
* For the coverage based approach to hierarchy, I think you missed an important early reference [1].
* I was confused for a while between the \phi and \psi functions. I would change one of these symbols to a more different one for clarity.
* L144: why do you estimate distance based on the value function? You have just fitted the adjancency network right, why do you not use this here?
* Alg 1: Is the subgoal transition function h() learned, or given? It seems to be given, but how is it specified?
* Alg 1: on what loss is the lower level trained? Just a standard goal-conditioned value-based loss?
* Sec 4.3: I found this quite hard to directly understand. I would first explain the main intuition, that you will construct a loss on the high-level policy to direct the direction in which it will sample a subgoal. Also, why do you not use the raw landmark as a subgoal, or the raw interpolation?
* The evaluation environments are rather small.
* Fig 4a: I do not see a difference between 40 and 60 really? Only 20 does not work?
* L241-253: And the other way around, what does coverage add over novelty?
* There are some typo’s in your document, double words, etc.
* There is no report on the run times of generating the landmark trees and running the planning algorithm. In the discussion you do mention that heavy planning budgets are allowed, but still, I would prefer if you reported on them.

Conclusion:
In general I think this is a decent submission. The topic is relevant: HRL, in particular the way to compress the higher level goal space. The authors extend previous work with a new way to estimate landmarks based on a combination of coverage and novelty. This seems to work well from their results, which is interesting. The integration of these landmarks in the HRL training loop is probably quite computationally heavy (the authors do not report on run times, which I consider one of the bigger issues with this paper), but does show good results (although the environments are rather simple, U-mazes only). Nevertheless, I do think this paper has value to the community, especially in the way the authors generate the landmarks.

**Time Spent Reviewing:**

1.5

---

> ### Author Response · Authors · 2021-08-10
> **Response to reviewer RGnx**
>
> Dear reviewer RGnx,
>
> We sincerely appreciate your valuable and insightful comments. We found them extremely helpful for improving our manuscript. We address each comment in detail, one by one below.
>
> ---
>
> **Q1. Suggestion in writing**
>
> **A1.** Thanks for helpful suggestions to improve clarity. We will fully incorporate suggestions in writing, including (1) more detailed explanations of adjacency networks and random network distillation in preliminaries, (2) additional references, and (3) changing notations for readability.
>
> ---
>
> **Q2. Distance estimation based on value function instead of adjacency network**
>
> **A2.** For distance estimation in building landmark-graph, we employ the value function instead of the adjacency network since the adjacency network is originally designed as a discriminator (which determines the adjacency of two states) rather than an estimator of distance. Nevertheless, one can adopt the adjacency network, but we expect that using the value function would achieve better performance.
>
> ---
>
> **Q3. Clarification on subgoal transition function**
>
> **A3.** In our paper, the subgoal transition function is “given” following prior works [1, 2]. It is specified as:
>
> $h(g_{t-1}, s_{t-1}, s_{t}) = g_{t-1} + s_{t-1} - s_{t},$
>
> The subgoal transition function is for updating a subgoal in a “relative” subgoal scheme. In the scheme, the high-level policy gives a subgoal that represents how far the low-level controller should move from its current state. As the current state is updated, the subgoal should be updated correspondingly. We will clarify this in the final manuscript.
>
> ---
>
> **Q4. Loss for training low-level policy**
>
> **A4.** We train the low-level policy with TD3 loss [3] (goal-conditioned value-based loss) as prior works [1, 2] did. We will clarify this in the final draft.
>
> ---
>
> **Q5. Why not use the raw landmark, or the raw interpolation rather than pseudo-landmark?**
>
> **A5.** Thanks for the suggestion. We will improve writing to raise understanding in Sec 4.3.
>
> We use pseudo-landmarks instead of “raw” selected landmarks in order to encourage generated subgoals to have both of desired properties: (1) “reachable” from the current state and (2) near promising states to explore (Note that the “raw” selected landmark only have the latter property). When a selected landmark is located too far from the current state, using “raw” landmarks would be suboptimal since it encourages high-level policy to generate unreachable subgoals from the current state. Instead, by introducing pseudo-landmarks, we can effectively ensure the high-level policy generates a subgoal that is (1) located near the current state and (2) toward the selected landmark. To verify the effectiveness of pseudo-landmarks, we additionally conduct ablative experiments with raw and pseudo landmarks, where it is more likely for landmarks to be located far from each other, i.e., Ant Maze (W-shape). We observe that using pseudo-landmarks achieves significantly better performance than using raw selected landmarks. We will incorporate the result in the final draft.
>
> ###### $\textbf{Success rate on Ant Maze (W-shape, sparse)}$
> \begin{array}{lcccc}
> \text{Timesteps}  & 2.5 * 10^5 & 5.0 * 10^5 & 7.5 * 10^5 & 10.0 * 10^5 \newline
> \hline
> \text{HIGL with \bf{pseudo-landmarks}}      & \bf{18.7 \\%} & \bf{30.2 \\%} & \bf{49.9 \\%} & \bf{52.6 \\%} \newline
> \text{HIGL with \bf{raw landmarks}}           & 14.6 \\%         & 27.2 \\%       & 32.0 \\%        & 42.4 \\% \newline
> \end{array}
>
> ---
>
> **Q6. The evaluation environment is rather small.**
>
> **A6.** To address your concern, we additionally evaluate HIGL over HRAC on more various tasks (i.e., robotic arm tasks and stochastic Ant Maze), and we observe that HIGL significantly surpasses HRAC across all the additional tasks. We will incorporate these results in the final draft.
>
> * We additionally evaluate HIGL over HRAC in robotic arm tasks (i.e, Reacher and Pusher) [4] and observe that HIGL outperforms HRAC. In the Reacher task, the objective is to manipulate the robotic arm to make the arm’s end-effector reach the goal position. In the Pusher task, the objective is to make a (puck-shaped) object in a plane reach a goal position by pushing the object using a robotic arm.
> ###### $\textbf{Success rate on Reacher}$
> \begin{array}{lcccc}
> \text{Timesteps}  & 2.0 * 10^5 & 3.0 * 10^5 & 4.0 * 10^5 & 5.0 * 10^5 \newline
> \hline
> \text{\textbf{HIGL}}      & \bf{39.1 \\%} & \bf{57.9 \\%} & \bf{73.2 \\%} & \bf{80.1 \\%} \newline
> \text{HRAC}      & 13.0 \\% & 9.5 \\% & 4.1 \\% & 7.4 \\% \newline
> \end{array}
> ###### $\textbf{Success rate on Pusher}$
> \begin{array}{lcccc}
> \text{Timesteps}  & 2.0 * 10^5 & 3.0 * 10^5 & 4.0 * 10^5 & 5.0 * 10^5 \newline
> \hline
> \text{\textbf{HIGL}}      & \bf{45.3 \\%} & \bf{50.6 \\%} & \bf{42.0 \\%} & \bf{48.6 \\%} \newline
> \text{HRAC}       & 6.0 \\% & 3.4 \\% & 2.8 \\% & 4.6 \\% \newline
> \end{array}
>
> * For stochastic environments, we employ the stochastic Ant Maze (U-shape) following set-ups from the prior work [1]. In stochastic Ant Maze (U-shape), gaussian noise with standard deviation $\sigma$ is added to the $(x, y)$ position of the ant robot at every step. As the table shows, HIGL shows significant gains over HRAC also in stochastic environments.
> ###### $\textbf{Success rate on Stochastic Ant Maze (U-shape, dense, $\sigma$=0.1) }$
> \begin{array}{lcccc}
> \text{Timesteps}  & 7.0 * 10^5 & 8.0 * 10^5 & 9.0 * 10^5 & 10.0 * 10^5 \newline
> \hline
> \textbf{HIGL}          & 0.8 \\% & \bf{5.2 \\%} & \bf{29.0 \\%}  & \bf{32.3 \\%}  \newline
> \text{HRAC}        & \bf{1.9 \\%} & 4.4 \\%  & 7.3 \\%  & 15.6 \\%  \newline
> \end{array}
>
> ---
> **Q7. In Figure 4a, no difference between 40 and 60 really? only 20 does not work?**
>
> **A7.** In Figure 4a, we understand that the marginal difference in performance between the number of landmarks of 40 and 60 comes from the fact that the pseudo-landmark can improve performance even when the number is not large (i.e., 40). We understand setting the number to 20 is too small to contain enough information to explore. In order to further raise the understanding tendency of the number of landmarks, we additionally provide experimental results of 30. We will incorporate the results of 30 in the final manuscript.
>
> ###### $\textbf{Success rate on Ant Maze (U-shape, dense)}$
> \begin{array}{lcccc}
> \text{Timesteps}  & 2.0 * 10^5 & 3.0 * 10^5 & 4.0 * 10^5 & 5.0 * 10^5 \newline
> \hline
> \text{HIGL with $M=\bf{60}$}      & 0.0 \\% & 11.1 \\% & \bf{44.6} \\% & \bf{64.0} \\% \newline
> \text{HIGL with $M=\bf{40}$}      & \bf{0.5} \\% & \bf{17.3} \\% & 35.7 \\% & 53.6 \\% \newline
> \text{HIGL with $M=\bf{30}$}      & 0.0 \\% & 1.4 \\% & 10.2 \\% & 22.2 \\% \newline
> \text{HIGL with $M=\bf{20}$}      & 0.0 \\% & 0.0 \\% & 0.0 \\% & 0.5 \\% \newline
> \end{array}
>
> ---
>
> **Q8. Suggested ablation study: Add coverage-based landmarks over novelty-based ones**
>
> **A8.** Coverage-based landmarks play an important role as a waypoint toward novel states or even as a promising state itself (See Figure 6 in the original draft for supporting qualitative analysis.) Following your suggestion, we provide additional experimental results with and without coverage-based landmarks to verify the effectiveness of coverage-based landmarks. As shown in the table, using coverage-based landmarks makes a significant gain. We will include the results and discussion in the final draft.
>
> ###### $\textbf{Success rate on Ant Maze (U-shape, dense)}$
> \begin{array}{lcccc}
> \text{Timesteps}  & 2.5 * 10^5 & 5.0 * 10^5 & 7.5 * 10^5 & 10.0 * 10^5 \newline
> \hline
> \text{HIGL with $M\_{\texttt{cov}} = 20$}      & \bf{1.4 \\%} & \bf{53.6 \\%} & \bf{80.7} \\% & \bf{88.6 \\%} \newline
> \text{HIGL with $M\_{\texttt{cov}} = 0$}      & 0.0 \\% & 1.4 \\% & 31.7 \\% & 58.4 \\% \newline
> \end{array}
>
> ---
>
> **Q9. Computational complexity**
>
> **A9.** We would like to note two facts about the computation complexity of HIGL; (a) HIGL is much more sample-efficient than HRAC, and (b) HIGL does not require planning in deployment time; hence the time consumed during deployment is the same as HRAC. Meanwhile, in training time, HIGL takes x2.1 wall-clock time more than HRAC. To be specific, for 10*10^5 training timesteps, HIGL takes 13 hours, and HRAC takes 6 hours using a single GPU (NVIDIA TITAN Xp) and 8 CPU cores (Intel Xeon E5-2630 v4). This is due to additional planning costs spent on training.
>
> ---
>
> **Reference**
>
> [1] Zhang, Tianren, et al. Generating adjacency- constrained subgoals in hierarchical reinforcement learning. Advances in Neural Information Processing Systems, 33, 2020.
>
> [2] Nachum, Ofir, et al. Data-Efficient Hierarchical Reinforcement Learning. Advances in Neural Information Processing Systems 31. 2018.
>
> [3] Fujimoto, Scott, Hoof, Herke, and Meger, David. Addressing function approximation error in actor-critic methods. In International Conference on Machine Learning, pp. 1587–1596. PMLR, 2018.
>
> [4] Chua, Kurtland, et al. Deep Reinforcement Learning in a Handful of Trials using Probabilistic Dynamics Models. Advances in Neural Information Processing Systems 31. 2018.

---

### Official Review · Reviewer_HpCw · 2021-07-15

**Rating:** 7
**Confidence:** 4

**Summary:**

This paper presents a new algorithm for goal-conditioned HRL, named HIGL, that trains a high-level policy to generate subgoals towards landmarks, i.e., promising states to explore. Two criteria are considered for being a landmark, coverage of the visited state space and novelty of states. Unlike prior work that only considered generating subgoals close to the current state, this work takes both (1) reachability and (2) potential information of subgoals into account. The proposed HIGL outperforms the previous state-of-the-art method in both dense and sparse reward settings thanks to the directed landmark-driven exploration by the high-level policy.


**Limitations And Societal Impact:**

The authors have discussed adequately limitation and social impact in Section 6.

**Main Review:**

Goal-conditioned HRL often suffers from training inefficiency because the high-level goal space can be as large as the state space. To address this well-known issue, the paper proposes a novel framework that conducts directed and landmark-driven exploration at the high-level to improve efficiency by leveraging a set of successful techniques in the literature. The idea is very intuitive, i.e., guiding the agent to promising states by generating reachable subgoals towards those states, but the actual implementation is nontrivial. The experimental results look quite promising, and I like the comparison with HRAC on sparse rewards, showcasing the importance of directed exploration.

Overall, the methodology is technically sound and the paper is well-written and organized. I am generally satisfied with the paper and enjoyed reading it. Nevertheless, I listed several my suggestions below (ordered by importance) that could further improve the paper if addressed or fully discussed.

(1)	More experiments: I encourage the authors to evaluate and compare HIGL against HRAC on more environments, such as Key-Chest, Ant Gather (both included in the HRAC paper) or any new environment that requires “directed/guided exploration”. In addition, can HIGL handle stochastic env. well?

(2)	Sparse env. results: (a) on Point Maze (sparse), why HRAC and HIGL achieved very similar performance? (HRAC slightly better) Is it because the size of the map is small compared to Ant Maze? (b) on both Ant Maze (sparse), the performance of HIGL fluctuated a lot during training and even dropped by the end of learning. Any thoughts on why this happened?

(3)	Parameter selection: the performance of HIGL varies with different parameter settings as shown in the Ablation studies. In practice, how to select these parameters or a reasonable combination of parameters (say for a new environment)? It seems to me that $\delta_{pseudo}$ is a key parameter, can the same $\delta_{pseudo}$ be used across different env.? In addition, how well HIGL can perform if using directly the selected landmark $l_t^{sel}$ instead of the pseudo-landmark in training the high-level? I would expect to see a performance drop but it can be added to the Ablation studies.

(4)	Novelty-based sampling: (line 132-133) the current approach discards old states based on the L2 norm in the goal space. What if the states are discarded based on the shortest transition distance, e.g., $\hat{d}_{st}(s, s’) < \lambda$? Will this affect the performance?

(5)	Computation complexity: it would be interesting to compare the computation time (HIGL vs. HRAC), so the readers understand how much computational overhead is needed.

(6)	High-dimensional imagery inputs: any ideas on how to generalize the framework to handle image inputs or image-based goals? This could be another limitation?

Minor comments:

(line 90-91) should eqn. (2) be $-||g_t - \phi(s_{t+1}) ||_2$? I would suggest using $||a-b||_2$ for Euclidean distance as in eqn. (3). Same applies to eqn. (4) to be consistent in notations.


**Time Spent Reviewing:**

3.25

---

> ### Author Response · Authors · 2021-08-10
> **Response to reviewer HpCw (1/2)**
>
> Dear reviewer HpCw
>
> We sincerely appreciate your valuable and insightful comments. We found them extremely helpful for improving our manuscript. We address each comment in detail, one by one below.
>
> ---
>
> **Q1. More environments including stochastic ones**
>
> **A1.** To address your concern, we additionally evaluate HIGL over HRAC on stochastic environments and complex tasks which require directed/guided exploration (i.e., robotic arm tasks). We observe that HIGL significantly surpasses HRAC over all the additional tasks. We will add these results to the final draft.
>
> * For a stochastic environment, we employ the stochastic Ant Maze (U-shape, dense) following the setup in the prior work [1]. In stochastic Ant Maze (U-shape, dense), Gaussian noise with standard deviation $\sigma$ is added to the $(x, y)$ position of the ant robot at every step. As the table shows, HIGL also outperforms HRAC significantly in the stochastic environment.
> ###### $\textbf{Success rate on Stochastic Ant Maze (U-shape, dense, $\sigma$=0.1) }$
> \begin{array}{lcccc}
> \text{Timesteps}  & 7.0 * 10^5 & 8.0 * 10^5 & 9.0 * 10^5 & 10.0 * 10^5 \newline
> \hline
> \textbf{HIGL}          & 0.8 \\% & \bf{5.2 \\%} & \bf{29.0 \\%}  & \bf{32.3 \\%}  \newline
> \text{HRAC}        & \bf{1.9 \\%} & 4.4 \\%  & 7.3 \\%  & 15.6 \\%  \newline
> \end{array}
>
> * Moreover, we additionally evaluate HIGL over HRAC in complex robotic arm tasks (i.e, Reacher and Pusher) [2], which require directed/guided exploration. We observe that HIGL significantly surpasses HRAC. In the Reacher task, the objective is to manipulate the robotic arm to make the arm’s end-effector reach the goal position. In the Pusher task, the objective is to make a (puck-shaped) object in a plane reach a goal position by pushing the object using a robotic arm.
> ###### $\textbf{Success rate on Reacher}$
> \begin{array}{lcccc}
> \text{Timesteps}  & 2.0 * 10^5 & 3.0 * 10^5 & 4.0 * 10^5 & 5.0 * 10^5 \newline
> \hline
> \text{\textbf{HIGL}}      & \bf{39.1 \\%} & \bf{57.9 \\%} & \bf{73.2 \\%} & \bf{80.1 \\%} \newline
> \text{HRAC}      & 13.0 \\% & 9.5 \\% & 4.1 \\% & 7.4 \\% \newline
> \end{array}
> ###### $\textbf{Success rate on Pusher}$
> \begin{array}{lcccc}
> \text{Timesteps}  & 2.0 * 10^5 & 3.0 * 10^5 & 4.0 * 10^5 & 5.0 * 10^5 \newline
> \hline
> \text{\textbf{HIGL}}      & \bf{45.3 \\%} & \bf{50.6 \\%} & \bf{42.0 \\%} & \bf{48.6 \\%} \newline
> \text{HRAC}       & 6.0 \\% & 3.4 \\% & 2.8 \\% & 4.6 \\% \newline
> \end{array}
>
> ---
>
> **Q2. More explanations about results on sparse environments**
>
> **A2.** We understand that HIGL is very effective in hard-exploration tasks thanks to its efficient exploration guided by landmarks. The relatively marginal gain in Point Maze is because the task is quite easier than Ant maze. The maze size of Point Maze is the same as that of Ant Maze (U-shape), but controlling a point robot is much easier (i.e, the dimension of low-level action space is 2, whereas 8 for an ant robot). We would like to emphasize that HIGL surpasses HRAC with a large gain in more difficult tasks of which directed exploration is important (i.e., Ant Maze (U-shape, W-shape)).
>
> For the fluctuation in HIGL, we understand that one reason would be the randomness that came from the coverage-based landmark sampling. For sampling coverage-based landmarks, we first build a pool of states via random sampling from a replay buffer and run the farthest point sampling in the pool. We suspect that the randomness in building a pool could cause fluctuation in training. It would be an interesting future direction to reduce fluctuation by improving schemes of building a pool.
>
> We understand that the drop by the end of training could come from the fixed number of landmarks even though visited state space becomes larger. In these situations, distances between landmarks may become further. Since distance estimation via value function is locally accurate but might be unreliable for far states, this may lead to selecting suboptimal landmarks in a landmark-graph. To verify this, we conduct additional experiments with an increased number of landmarks (from 80 to 120) to alleviate the inaccurate estimation when the visited state space enlarges. We observe that drop by the end of training is quite a lot removed if we use the increased number of landmarks.
>
> ###### $\textbf{Success rate on Ant Maze (W-shape, sparse)}$
> \begin{array}{lcccc}
> \text{Timesteps}  & 7.0 * 10^5 & 8.0 * 10^5 & 9.0 * 10^5 & 10.0 * 10^5 \newline
> \hline
> \text{HIGL with $M=\bf{80}$}      & 41.7 \\% & 36.1 \\% & 32.1 \\% & 34.5 \\% \newline
> \text{HIGL with $M=\bf{120}$}      & \bf{46.5 \\%} & \bf{58.3 \\%} & \bf{58.1 \\%} & \bf{60.6 \\%} \newline
> \end{array}
>
> ---
>
> **Q3. Hypyerparameter selection**
>
> **A3.** HIGL achieves superior performance over HRAC without an exhaustive search of hyperparameters, so we did not explore hyperparameters to a large extent (i.e., we use shift magnitude as 2.0 for all the four Ant Mazes, 0.5 for two Point Mazes in the original submission). Nevertheless, one can adjust hyperparameters (1) shift magnitude, (2) number of landmarks, and (3) adjacency degree by following reasonable guidelines (especially for a new environment), which will be added to the final draft.
>
> For (1) shift magnitude, performance would be further improved if one adjusts shift magnitude across different environments. Here, one important point is to set “balanced” shift magnitude; too large magnitude would make pseudo-landmarks unreachable, whereas too small magnitude makes no explorative benefits. To this end, for example, one can use the following choice of $\delta_{\texttt{pseudo}}$:
>
> $\delta_{\texttt{pseudo}} = \mathbb{E} \Vert g_{t}^{\texttt{sel}} - g_{t}^{\texttt{cur}} \Vert_{2}$
>
> Namely, it is the average of the distance between selected landmarks and the current state in the goal space. To verify the effectiveness of the automatic setting, we additionally conduct experiments and observe the superior performance over HRAC.
>
> ###### $\textbf{Success rate on Ant Maze (U-shape, dense)}$
> \begin{array}{lcccc}
> \text{Timesteps}  & 2.5 * 10^5 & 5.0 * 10^5 & 7.5 * 10^5 & 10.0 * 10^5 \newline
> \hline
> \textbf{HIGL with automatic shift magnitude}      & \bf{1.3 \\%} & \bf{59.9 \\%} & \bf{83.5 \\%} & \bf{82.5 \\%}  \newline
> \text{HRAC}        & 0.0 & 5.0 \\% & 47.9 \\% & 78.5 \\%  \newline
> \end{array}
>
> For (2) number of landmarks, the too small number could cause a performance drop, but we emphasize that our scheme of pseudo-landmarks could improve the performance given the same number of landmarks. To be specific, if the number of landmarks is not large, it is likely that selected landmarks are located far from the current state. Training high-level policy using such selected landmarks could be suboptimal since it encourages high-level policy to generate unreachable subgoals from the current state. Instead, we offer pseudo-landmarks that satisfy both desired properties: (a) reachable from the current state and (b) directed toward promising states. You can verify the effectiveness in the response of Question 4.
>
> For (3) adjacency degree, we suggest setting it slightly smaller than given high-level action frequency (we note that the adjacency degree is not sensitive to achieve superior performance over HRAC). Since the pseudo-landmark is located near the current state, setting an adjacency degree such a way could make the region near the pseudo-landmarks “reachable” from the current state. Note that setting an adjacency degree too large could allow even unreachable regions into consideration when training high-level policy, which is undesirable.
>
> ---
>
> **Q4. how does HIGL perform when the selected landmarks are directly used instead of the pseudo-landmarks for training a high-level policy?**
>
> **A4.** Following your comment, we perform HIGL using the selected landmark instead of the pseudo-landmark in training the high-level policy. As you expected, we observe a performance drop, which demonstrates the effectiveness of using pseudo-landmarks. We will add the results in the ablation studies.
>
> ###### $\textbf{Success rate on Ant Maze (W-shape, sparse)}$
> \begin{array}{lcccc}
> \text{Timesteps}  & 2.5 * 10^5 & 5.0 * 10^5 & 7.5 * 10^5 & 10.0 * 10^5 \newline
> \hline
> \text{HIGL with \bf{pseudo-landmarks}}      & \bf{18.7 \\%} & \bf{30.2 \\%} & \bf{49.9 \\%} & \bf{52.6 \\%} \newline
> \text{HIGL with \bf{raw landmarks}}           & 14.6 \\%         & 27.2 \\%       & 32.0 \\%        & 42.4 \\% \newline
> \end{array}
>
> ---
>
> **Q5. Another design choice for discarding old states in the novelty priority queue**
>
> **A5.** Following your suggestion, we additionally conduct experiments with replacing our design choice of discarding old states in the novelty priority queue with your suggested alternative. For the alternative, we use the adjacency network to estimate the shortest transition distance. As shown in the table, our original design choice shows slightly better performance (note that regardless of discarding design choice, HIGL is much better than HRAC). Nevertheless, it would be an interesting research direction to improve discarding design in the novelty priority queue. We will incorporate these results in the final draft. Thank you!
>
> ###### $\textbf{Success rate on Ant Maze (U-shape, dense)}$
> \begin{array}{lcccc}
> \text{Timesteps}  & 2.5 * 10^5 & 5.0 * 10^5 & 7.5 * 10^5 & 10.0 * 10^5 \newline
> \hline
> \text{HIGL with \textbf{original} discarding design}      & 1.4 \\% & \bf{53.6 \\%} & \bf{80.7 \\%} & 88.6 \\% \newline
> \text{HIGL with \textbf{alternative} discarding design}      & \bf{6.7 \\%} & 40.4 \\% & 71.5 \\% & \bf{89.5 \\%} \newline
> \text{HRAC}      & 0.0 \\% & 5.0 \\% & 47.9 \\% & 78.5 \\% \newline
> \end{array}

---

> > ### Author Response · Authors · 2021-08-10
> > **Response to reviewer HpCw (2/2)**
> >
> > **Q6. Computation complexity**
> >
> > **A6.** We would like to note two facts about the computation complexity of HIGL; (a) HIGL is much more sample-efficient than HRAC, and (b) HIGL does not require planning in deployment time; hence the time consumed during deployment is the same as HRAC. Meanwhile, in training time, HIGL takes x2.1 wall-clock time more than HRAC. To be specific, for 10*10^5 training timesteps, HIGL takes 13 hours, and HRAC takes 6 hours using a single GPU (NVIDIA TITAN Xp) and 8 CPU cores (Intel Xeon E5-2630 v4). This is due to additional planning costs spent on training.
> >
> > ---
> >
> > **Q7. How to generalize HIGL to handle image inputs or imaged-based goals?**
> >
> > **A7.** In principle, HIGL is applicable to environments with high-dimensional state spaces, and we think it would be interesting to explore in the future. A potential issue in such environments is that the required number of landmarks would be increased, but we expect that it can be efficiently reduced by combining subgoal representation learning (orthogonal methodology to HIGL). An increased number of landmarks can lead to spending more time in planning over a landmark-graph. To alleviate this issue, one can build the priority queue and the landmark-graph using landmarks in “goal space” instead of “state space”; goal space typically has a lower dimension. The reason why one can build them in “goal space” comes from the fact that HIGL eventually utilizes landmarks in “goal space” rather than “state space”, as equation (7) in the original draft shows:
> >
> > $\mathcal{L}\_{\texttt{landmark}}(\theta\_{\texttt{high}}) = \max(|| \psi\_{\phi}(g\_{t}^{\texttt{pseudo}}) - \psi\_{\phi}(g\_{t})||_{2} - \varepsilon\_{k}, 0)$
> >
> > We expect that HIGL combined with subgoal representation learning (which learns state to goal mapping function) would be successful since it has shown promising performance on environments with high-dimensional state spaces [3, 4]. We will discuss this in the final draft.
> >
> > ---
> >
> > **Q8. Minor comments: reward shaping for low-level policy, suggestion in notation**
> >
> > **A8.** Thanks for the suggestion. We will improve notations in the equation. (3) and (4).
> >
> > The low-level reward is shaped in such a way in equation (2) since we employ a “relative” subgoal scheme for low-level policy, following prior works [1, 2]. In the scheme, the high-level policy gives a subgoal that represents how far the low-level controller should move from its current state. If an “absolute” subgoal scheme is used instead, one can replace it with the low-level reward shaping you commented. In the final draft, we will clarify the low-level reward shaping.
> >
> > ---
> >
> > **Reference**
> >
> > [1] Zhang, Tianren, et al. Generating adjacency- constrained subgoals in hierarchical reinforcement learning. Advances in Neural Information Processing Systems, 33, 2020.
> >
> > [2] Chua, Kurtland, et al. Deep Reinforcement Learning in a Handful of Trials using Probabilistic Dynamics Models. Advances in Neural Information Processing Systems 31. 2018.
> >
> > [3] Nachum, Ofir, et al. Near-Optimal Representation Learning for Hierarchical Reinforcement Learning. International Conference on Learning Representations. 2018.
> >
> > [4] Li, Siyuan, et al. Learning Subgoal Representations with Slow Dynamics. International Conference on Learning Representations. 2020.

---

> > > ### Comment · Reviewer_HpCw · 2021-08-30
> > > **Thank you**
> > >
> > > I appreciate the authors for their thorough and thoughtful response. All my questions/concerns have been adequately addressed, and I am impressed by the additional results on the robotic arm tasks. Please add the additional experiments and discussion in the final version, and I believe they are valuable and helpful for the readers.

---

> > > > ### Author Response · Authors · 2021-08-30
> > > > **Thank you for the response**
> > > >
> > > > We are happy to hear that our rebuttal addressed your concerns well.
> > > >
> > > > We will definitely add the additional results and discussion on the robotic arm tasks to the final version, which we are also quite excited with.
> > > >
> > > > Thank you again for the valuable suggestions and comments, which we believe further strengthen our paper !
> > > >
> > > > Best, Authors.

---

### Official Review · Reviewer_zAqr · 2021-07-30

**Rating:** 6
**Confidence:** 5

**Summary:**

The paper presents a novel hierarchical reinforcement learning guided by landmarks (HIGL), which can reduce the high-level action space and improve the training and exploration efficiency of hierarchical reinforcement learning (HRL). Empirical results demonstrated that HIGL outperformed one state-of-the-art HRL approach.


**Limitations And Societal Impact:**

The following questions need to be addressed.

Q.1 Although the work is well-motivated, it might be seen as a combination of the study by Zhang et al. [1] and the study by Huang et al. [2], which weakens the novelty to a large extent. Thus, I encourage the authors to theoretically analyze the advantages of HIGL to improve the paper's contribution.

References:

[1] Zhang, Tianren, Guo, Shangqi, Tan, Tian, Hu, Xiaolin, and Chen, Feng. Generating adjacency- constrained subgoals in hierarchical reinforcement learning. Advances in Neural Information Processing Systems, 33, 2020.

[2] Huang, Zhiao, Liu, Fangchen, and Su, Hao. Mapping state space using landmarks for universal goal reaching. Advances in Neural Information Processing Systems, 32:1942–1952, 2019.

Q.2 Some techniques, such as priority queue $\mathcal{Q}$ and a state graph, seem to be difficult to apply to high-dimensional state spaces and stochastic environments. I suspect it is necessary to evaluate HIGL over some stochastic environments.

Q.3 How to automatically set shift magnitude $\delta_{\rm pseudo}$? As shown in Figure 4(b), the HIGL performance is very sensitive to $\delta_{\rm pseudo}$. I believe it is critical to automatically set shift magnitude $\delta_{\rm pseudo}$.

Q.4 The results shown in Figures 3(e) and 3(f) are quite surprising. I would like to see more discussions about the reason why HIGL largely surpasses HRAC in the two tasks.

Q.5 Does HIGL suffer from performance degeneration when landmarks are insufficient? Do high-dimensional state spaces need a large number of landmarks?

**Main Review:**

The paper is well written, the motivation is quite clear, and the experiments are diversified. However, some critical issues need to be addressed before I recommend acceptance.


**Time Spent Reviewing:**

5.25

---

> ### Author Response · Authors · 2021-08-10
> **Response to reviewer zAqr**
>
> Dear reviewer zAqr,
>
> We sincerely appreciate your valuable and insightful comments. We found them extremely helpful for improving our manuscript. We address each comment in detail, one by one below.
>
> ---
>
> **Q1. Combining existing works lacking novelty**
>
> **A1.** We emphasize that HIGL is not the naive combination of the prior works [1, 2] you commented. Although we agree that HIGL adopted some components from them [1, 2], our novelty is on the overall pipeline of HIGL which is the first method that learns a high-level policy guided by planning in a graph. Moreover, as noted by (Reviewer HpCw), we have new and nontrivial algorithmic components (e.g., novelty-based sampling and pseudo-landmarks) not existing in [1, 2], which are demonstrated to be effective in various environments. As highlighted by (Reviewer RGnx), we believe that HIGL proposes a novel and interesting research direction to the HRL community.
>
> ---
>
> **Q2. Applicability to stochastic environments**
>
> **A2.** HIGL is applicable to stochastic environments without any modification since our algorithmic components (including the novelty priority queue and a landmark-graph) are built on visited states, regardless of transition dynamics. To empirically show that HIGL is effective on stochastic environments, we additionally provide experimental results on stochastic Ant Maze (U-shape, dense), where Gaussian noise $N(0, 0.1)$ is added to the $(x, y)$ position of the ant robot at every step, following the setup in [1]. As the table shows, HIGL also outperforms HRAC in stochastic environments. We will incorporate this result in the final manuscript.
>
> ###### $\textbf{Success rate on Stochastic Ant Maze (U-shape, dense, $\sigma$=0.1) }$
> \begin{array}{lcccc}
> \text{Timesteps}  & 7.0 * 10^5 & 8.0 * 10^5 & 9.0 * 10^5 & 10.0 * 10^5 \newline
> \hline
> \textbf{HIGL}          & 0.8 \\% & \bf{5.2 \\%} & \bf{29.0 \\%}  & \bf{32.3 \\%}  \newline
> \text{HRAC}        & \bf{1.9 \\%} & 4.4 \\%  & 7.3 \\%  & 15.6 \\%  \newline
> \end{array}
>
> ---
>
> **Q3. Applicability to an environment of high-dimensional state spaces. Do high-dimensional state spaces need a large number of landmarks?**
>
> **A3.** In principle, HIGL is applicable to environments with high-dimensional state spaces, and we think it would be interesting to explore in the future. A potential issue in such environments is that the required number of landmarks would be increased, but we expect that it can be efficiently reduced by combining subgoal representation learning (orthogonal methodology to HIGL). An increased number of landmarks can lead to spending more time in planning over a landmark-graph. To alleviate this issue, one can build the priority queue and the landmark-graph using landmarks in “goal space” instead of “state space”; goal space typically has a lower dimension. The reason why one can build them in “goal space” comes from the fact that HIGL eventually utilizes landmarks in “goal space” rather than “state space”, as equation (7) in the original draft shows:
>
> $\mathcal{L}\_{\texttt{landmark}}(\theta\_{\texttt{high}}) = \max(|| \psi\_{\phi}(g\_{t}^{\texttt{pseudo}}) - \psi\_{\phi}(g\_{t})||_{2} - \varepsilon\_{k}, 0)$
>
> We expect that HIGL combined with subgoal representation learning (which learns state to goal mapping function) would be successful since it has shown promising performance on environments with high-dimensional state spaces [3, 4]. We will discuss this in the final draft.
>
> ---
>
> **Q4. Automatic setting of shift magnitude $\delta_{\texttt{pseudo}}$**
>
> **A4.** We first remark that the superior performance of HIGL is achieved without exhaustive hyperparameter search on the shift magnitude $\delta_{\texttt{pseudo}}$ for each environment; we use $\delta_{\texttt{pseudo}}=2.0$ for all the four Ant Mazes, $\delta_{\texttt{pseudo}}=0.5$ for two Point Mazes in the original draft.
>
> Nevertheless, as you expected, one can improve the performance of HIGL by setting shift magnitude in a systematic manner. Here, one important point is to set “balanced” shift magnitude; too large magnitude would make pseudo-landmarks unreachable, whereas too small magnitude makes no explorative benefits. To this end, for example, one can use the following choice of $\delta_{\texttt{pseudo}}$:
>
> $\delta_{\texttt{pseudo}} = \mathbb{E} \Vert g_{t}^{\texttt{sel}} - g_{t}^{\texttt{cur}} \Vert_{2}$
>
> Namely, it is the average of the distance between selected landmarks and the current state in the goal space. To verify the effectiveness of the automatic setting, we additionally conduct experiments and observe the superior performance over HRAC. We will add this automatic selection strategy of $\delta_{\texttt{pseudo}}$ to the final draft. Thank you!
>
> ###### $\textbf{Success rate on Ant Maze (U-shape, dense)}$
> \begin{array}{lcccc}
> \text{Timesteps}  & 2.5 * 10^5 & 5.0 * 10^5 & 7.5 * 10^5 & 10.0 * 10^5 \newline
> \hline
> \textbf{HIGL with automatic shift magnitude}      & \bf{1.3 \\%} & \bf{59.9 \\%} & \bf{83.5 \\%} & \bf{82.5 \\%}  \newline
> \text{HRAC}        & 0.0 & 5.0 \\% & 47.9 \\% & 78.5 \\%  \newline
> \end{array}
>
> ---
>
> **Q5. More discussion about large gain by HIGL in the Ant Maze (U-shape, W-shape) with sparse reward.**
>
> **A5.** The significant gain of HIGL in tasks of sparse rewards (in Figures 3(e) and 3(f)) comes from the “efficient high-level exploration” guided by landmarks (promising states to explore). To be specific, instead of treating all the adjacent states equally (as HRAC did), HIGL considers both reachability and the novelty of a state. HIGL recognizes promising directions to explore via planning and trains high-level policy to generate a subgoal toward the direction. We understand that such differences in our mechanism made a large gain over HRAC.
>
> Furthermore, we would like to inform you that the “efficient high-level exploration” guided by landmarks also improves performance in complex tasks such as robotic arms tasks [5]. As shown in the tables below, HIGL surpasses HRAC across various complex tasks by a large margin. In the Reacher task, the objective is to manipulate the robotic arm to make the arm’s end-effector reach the goal position. In the Pusher task, the objective is to make a (puck-shaped) object in a plane reach a goal position by pushing the object using a robotic arm. We will add the results to the final draft.
>
> ###### $\textbf{Success rate on Reacher}$
> \begin{array}{lcccc}
> \text{Timesteps}  & 2.0 * 10^5 & 3.0 * 10^5 & 4.0 * 10^5 & 5.0 * 10^5 \newline
> \hline
> \text{\textbf{HIGL}}      & \bf{39.1 \\%} & \bf{57.9 \\%} & \bf{73.2 \\%} & \bf{80.1 \\%} \newline
> \text{HRAC}      & 13.0 \\% & 9.5 \\% & 4.1 \\% & 7.4 \\% \newline
> \end{array}
>
> ###### $\textbf{Success rate on Pusher}$
> \begin{array}{lcccc}
> \text{Timesteps}  & 2.0 * 10^5 & 3.0 * 10^5 & 4.0 * 10^5 & 5.0 * 10^5 \newline
> \hline
> \text{\textbf{HIGL}}      & \bf{45.3 \\%} & \bf{50.6 \\%} & \bf{42.0 \\%} & \bf{48.6 \\%} \newline
> \text{HRAC}       & 6.0 \\% & 3.4 \\% & 2.8 \\% & 4.6 \\% \newline
> \end{array}
>
> ---
>
> **Q6. Does HIGL suffer from performance degradation when landmarks are insufficient?**
>
> **A6.** HIGL may suffer from performance degradation when landmarks are insufficient, but we emphasize that our scheme of pseudo-landmarks could alleviate the issue. If the number of landmarks is not sufficient, two issues can occur; (1) it would be hard to contain enough information for high-level exploration, (2) selected landmarks are likely to be located too far from the current state. However, we note that our scheme of generating pseudo-landmarks can alleviate the issue (2) a lot. Using such (far) selected landmarks is undesirable since it encourages high-level policy to generate unreachable subgoals from the current state. Instead, for training high-level policy, we provide pseudo-landmarks that satisfy both desired properties: (a) reachable from the current state and (b) directed toward promising states. By using pseudo-landmarks, we can improve the performance of HIGL given the same number of landmarks. To verify this, we additionally compare the performance with and without pseudo-landmarks, given the same number of landmarks. We will add the results of HIGL with pseudo-landmarks to the final draft.
>
> ###### $\textbf{Success rate on Ant Maze (W-shape, sparse)}$
> \begin{array}{lcccc}
> \text{Timesteps}  & 2.5 * 10^5 & 5.0 * 10^5 & 7.5 * 10^5 & 10.0 * 10^5 \newline
> \hline
> \text{HIGL with \bf{pseudo-landmarks}}      & \bf{18.7 \\%} & \bf{30.2 \\%} & \bf{49.9 \\%} & \bf{52.6 \\%} \newline
> \text{HIGL with \bf{raw landmarks}}           & 14.6 \\%         & 27.2 \\%       & 32.0 \\%        & 42.4 \\% \newline
> \end{array}
>
> ---
>
> **Reference**
>
> [1] Zhang, Tianren, et al. Generating adjacency- constrained subgoals in hierarchical reinforcement learning. Advances in Neural Information Processing Systems, 33, 2020.
>
> [2] Chua, Kurtland, et al. Deep Reinforcement Learning in a Handful of Trials using Probabilistic Dynamics Models. Advances in Neural Information Processing Systems 31. 2018.
>
> [3] Nachum, Ofir, et al. Near-Optimal Representation Learning for Hierarchical Reinforcement Learning. International Conference on Learning Representations. 2018.
>
> [4] Li, Siyuan, et al. Learning Subgoal Representations with Slow Dynamics. International Conference on Learning Representations. 2020.

---

> > ### Comment · Reviewer_zAqr · 2021-08-10
> > **One More Question about Training Steps**
> >
> > Thank you very much for the response. I appreciate the effort that the authors put into addressing my questions. I believe that the above analysis and discussion can significantly improve the quality of the manuscript. Overall, I am tending to raise my score.
> >
> > Before that, I have one more question. The authors trained HIGL and HRAC with only 1M training steps, but Zhang et al. trained HRAC with total 5M training steps over Ant Maze tasks. To make a fair comparison, I think it is necessary to compare the performance of HRAC and HIGL during 5M training steps.

---

> > > ### Author Response · Authors · 2021-08-16
> > > **Response to reviewer zAqr (2)**
> > >
> > > Thanks again for your comments. We choose 1M timesteps since it is enough for HIGL to achieve a near success rate of 90% under the tested Ant Maze (U-shape) of size $\bf{12\*12}$ [1]
> > > (note that the size of Ant Maze (U-shape) in HRAC paper [2] is $\bf{24\*24}$, i.e, larger than that in [1]). Nevertheless, to further address your concern, we additionally compare HIGL and HRAC under the same setups in [2], i.e., 5M timesteps in $\bf{24\*24}$ Ant Maze (U-shape) and $\bf{20\*20}$ Ant Maze (W-shape). The tables below show that HIGL is much more sample-efficient than HRAC, while achieving a similar asymptotic performance, which clearly shows the benefit of the proposed landmark-guided exploration. We will include the relevant experimental results and discussion in the final draft. Please let us know if you have any more questions or concerns.
> > >
> > > ###### $\textbf{Success rate on Ant Maze (U-shape, dense) of size 24 \* 24}$
> > > \begin{array}{lcccc}
> > > \text{Timesteps}  & 2.5 * 10^5 & 5.0 * 10^5 & 7.5 * 10^5 & 10.0 * 10^5 & 20.0 * 10^5 & 30.0 * 10^5 & 40.0 * 10^5 & 50.0 * 10^5 \newline
> > > \hline
> > > \textbf{HIGL}       & 0.0 \\% & \bf{11.5} \\% & \bf{53.3} \\% & \bf{70.0} \\% & \bf{82.7} \\% & \bf{80.8} \\% & \bf{84.7} \\% & 77.5 \\% \newline
> > > \text{HRAC}      & 0.0 \\% & 0.0 \\% & 17.6 \\% & 55.7 \\% & 75.5 \\% & 77.7 \\% & 83.6 \\% & \bf{82.6} \\% \newline
> > > \end{array}
> > >
> > > ###### $\textbf{Success rate on Ant Maze (W-shape, dense) of size 20 \* 20}$
> > > \begin{array}{lcccc}
> > > \text{Timesteps}  & 2.5 * 10^5 & 5.0 * 10^5 & 7.5 * 10^5 & 10.0 * 10^5 & 20.0 * 10^5 & 30.0 * 10^5 & 40.0 * 10^5 & 50.0 * 10^5 \newline
> > > \hline
> > > \textbf{HIGL}       & \bf{29.6} \\%  & \bf{63.4} \\% & \bf{73.7} \\% & \bf{78.7} \\% & \bf{70.4} \\% & \bf{80.5} \\% & \bf{71.6} \\% & 82.0 \\%\newline
> > > \text{HRAC}      & 15.4 \\%  & 35.3 \\% & 38.2 \\% & 46.8 \\% & 40.1 \\% & 61.2 \\% & 63.6 \\% & \bf{83.5} \\%\newline
> > > \end{array}
> > >
> > > ---
> > >
> > > **Reference**
> > >
> > >
> > > [1] Huang, Zhiao, et al. Mapping state space using landmarks for universal goal reaching. Advances in Neural Information Processing Systems, 32:1942–1952, 2019.
> > >
> > > [2] Zhang, Tianren, et al. Generating adjacency- constrained subgoals in hierarchical reinforcement learning. Advances in Neural Information Processing Systems, 33, 2020.

---

> > > > ### Comment · Reviewer_zAqr · 2021-08-17
> > > > **My concern has been addressed**
> > > >
> > > > Thank you for the response that addresses my concern. Please add the experimental results and discussion into the final version. I would like to raise the score.

---

> > > > > ### Author Response · Authors · 2021-08-30
> > > > > **Thank you for the response**
> > > > >
> > > > > We are happy to hear that our rebuttal addressed your concerns well.
> > > > >
> > > > > We will add the suggested experimental results and discussion into the final version.
> > > > >
> > > > > Thank you again for the valuable suggestions and comments, which we believe further strengthen our paper !
> > > > >
> > > > > Best, Authors.

---

### Decision · Program_Chairs · 2021-09-27

**Decision:**

Accept (Poster)

**Comment:**

Reviewers found the paper to contain novel ideas with solid justifications. The author response was particularly useful in addressing some of the initially raised issues. We therefore recommend acceptance.

The authors are encouraged to take the reviews into account to improve presentation and results, especially to include some of the new ones that were provided in the rebuttal. I’d also strongly encourage the authors to strengthen the related work section. The paper unfortunately ignores over two decades of significant hierarchical RL work, some of which appear related enough to deserve a discussion. Examples are the following subgoal discovery work that rely on graph structures, and some of them have a notion of novelty (although different from this work):
https://dl.acm.org/doi/abs/10.1007/3-540-36755-1_25
https://dl.acm.org/doi/abs/10.1145/1015330.1015355
https://dl.acm.org/doi/abs/10.1145/1015330.1015353

A potential challenge mentioned by the reviewers is to scale to high-dimensional space, and the authors suggested working in the goal space. The following paper works in the goal space, and relies on successful trajectories to find goals in high-dimensional state spaces: https://aclanthology.org/D18-1253 (https://aclanthology.org/D18-1253/).